# Assessing flooding impact to riverine bridges: an integrated analysis

Maria Pregnolato[1*,] Andrew O. Winter[2], Dakota Mascarenas[2], Andrew D. Sen[3], Paul Bates[4], Michael R. Motley[2]

[1] Dep. of Civil Engineering, University of Bristol, Bristol, BS8 1TR, UK
[2] Dep. of Civil and Environmental Engineering, University of Washington, Seattle, 98103, USA
[3] Dep. of Civil, Construction and Environmental Engineering, Marquette University, Milwaukee, 53233, USA
[4] School of Geographical Sciences, University of Bristol, Bristol, BS8 1RL, UK

*Correspondence to*: Maria Pregnolato (maria.pregnolato@bristol.ac.uk)

**Abstract.** Flood events are the most frequent cause of damage to infrastructure compared to any other natural hazard, and global changes (climate, socio-economic, technological) are likely to increase this damage. Transportation infrastructure systems are responsible for moving people, goods, and services, and ensuring connection within and among urban areas. A failed link in this system can impact the community by threatening evacuation capability, recovery operations and the overall economy. Bridges are critical links in the wider urban system since they are associated with little redundancy and a high (re)construction cost. Riverine bridges are particularly prone to failure during flood events; in fact, the risks to bridges from high river flows and bank erosion have been recognized as crucial at global level. The interaction among flow, structure and network is complex, and not fully understood. This study aims to establish a rigorous, multiphysics modelling approach for the assessment of the hydrodynamic forces impacting inundated bridges, and the subsequent structural response, while understanding the consequences of such impact on the surrounding network. Objectives of this study are to model hydrodynamic forces as demand on the bridge structure, to advance a performance evaluation of the structure under the modelled loading, and to assess the overall impact at systemic level. The flood-prone city of Carlisle (UK) is used as case study and a proof of concept. Implications of the hydrodynamic impact on the performance and functionality of the surrounding transport network are discussed. This research will help to fill the gap between current guidance for design and assessment of bridges within the overall transport system.

## 1 Introduction

Bridges are crucial elements of the transport network given their high construction costs and the lack of alternatives routes. Man-made and natural events are a threat to bridge safety and network serviceability (Yang and Frangopol, 2020). Bridges act as bottlenecks for surrounding roads, and thus any service disruption can knock-out communities' access and connections, impair emergency planning and evacuation routes, as well as impact economies and businesses.

Some disruptive events are growing in frequency and severity. In particular, the impacts of flooding have been exacerbated in recent years by urbanisation (e.g. increase of impermeable surfaces), inappropriate land use in flood-prone areas and climate change. Rainfall events that lead to flooding are becoming more frequent and intense (Solomon et al., 2007), triggering bridge incidents and failures all over the world (Cumbria, UK, 2009; Drake, Colorado, 2013; Texas, 2018; Greece, 2020). As recent examples, Grinton Bridge in Yorkshire (North-West UK) and Keritis Bridge in Crete (Greece) were both washed away by floodwaters in 2019.

Riverine bridges are intrinsically vulnerable to flooding, as they are located in the area of the riverbed. Flood and scour represent one of the most frequent causes of bridge failures (Hunt, 2009; Wardhana and Hadipriono, 2003; Khan, 2015; Ahamed et al., 2020). Although, scour is recognized as the biggest threat for bridges over water (and available scour-related literature is much more robust), hydrodynamic forces could be as critical for bridge piers on bedrock (where scour is unlikely), and for the decks of all flooded bridges (Kim et al., 2017; Oudenbroek et al., 2018). In terms of consequences, natural hazards can damage bridges structurally (thus causing direct physical damages), but these events can also result in

functional failures that cause travel time delays and rerouting that lead to indirect losses (Alabbad et al., 2021). Any bridge
failure, whether structural or functional, has the potential to impose heavy consequences to owners or responsible authorities,
as well as dire expenses. Therefore, understanding the potential impact of flooding to bridges is a compelling need of
communities in areas of high flood risk.
Currently, a limited number of studies investigated the consequences of extreme flooding to bridges and the surrounding
network (Yang and Frangopol, 2020). Practical application and case studies of real bridges tend to be focused on other
natural hazards (e.g. earthquakes: Kilanitis and Sextos, 2019, Ertugay et al., 2016; Zhou et al., 2010). This study aims to
establish a rigorous, multiphysics modelling approach for assessing hydrodynamic forces on inundated bridges, subsequent
structural response, and consequences of such impact on the surrounding network. Objectives of this study are to model
hydrodynamic forces as demand on the bridge structure, to advance a performance evaluation of the structure under the
modelled loading, and to assess the overall impact at systemic level. Implications of the hydrodynamic impact on the
performance and functionality of the surrounding transport network are discussed. This research will help to fill the gap
between current guidance for design and assessment of bridges within the overall transport system.
**1.1 Background**
Transport networks are formed by multiple links (i.e. roads), and their performance relies on a number of parameters, such as
availability of alternative routes (redundancy), road capacity, or traffic demand, among others. A bridge failure often means
a critical link been taken out of service. Bridges are usually costly assets to be repaired, have little redundancy and are likely
to be crossed by a high number of users, especially if belonging to strategic road networks (e.g. highways). Therefore, bridge
closure or failure can impact the overall performance of the road network and the failure consequences have to be
investigated from a system-perspective (Yang and Frangopol, 2020). The assessment of the systemic impact is a complex
and multi-disciplinary problem, at the interface of hydrology, fluid dynamics, structural analysis and transport modelling.
Scour damage is a significant concern for many bridge structures and has been extensively studied (e.g. Pregnolato et al.,
2021a; Wang et al., 2017; Hung and Yau, 2017; AASHTO, 2002); the more common methods include using the HEC-18
(Arneson et al., 2012) or CIRIA scour equations (Kirby et al., 2015; HE, 2012). however, it is not the main focus of this
paper

On the contrary, literature about modeling the hydrodynamic forces of the fluid on bridges due to riverine floods is limited,
especially concerning fragility models or reliability analysis (Pregnolato, 2019; Gidaris et al., 2017). Existing research
investigated tsunami impact to bridges (e.g. Motley and al., 2016; Lomonaco et al., 2018; Qin et al., 2018; Winter et al.,
2017), where Computational Fluid Dynamics (CFD) techniques are used to compute hydrodynamic forces on bridges and
components. Li et al. (2021) advanced a CFD-based numerical study on the tsunami-induced scour around bridge piers.
Kerenyi et al. (2009) applied CFD to compute hydrodynamic forces on inundated bridge decks, however the analysis was
limited to the evaluation of drag and lift forces, without investigating impact and consequences. Bento et al., (2021)
suggested CFD as a more sophisticated technique for modelling flow depth and velocities at sites. Multi-hazard studies have
investigated the interaction and implication of multiple hazards acting on a single structure (Gidaris et al., 2017; Carey et al.,
2019), especially between earthquake and tsunami. Other studies (Mondoro and Frangopol, 2018; Liu et al., 2018; Yilmaz et
al., 2016) that tackled flood impact to bridges generally expressed the hazard through flood hazard curves, generated via
flood-frequency analysis; however, a detailed hydraulic analysis was beyond the scope of their work. While tsunami loading
of bridges will often result in much higher forces than riverine flows, the prevalence of riverine flooding relative to tsunami
events necessitate further study and could have a far-reaching effect.

## 1.2 Motivation and aim

To the authors' knowledge, no study has comprehensively investigated the impact of high-river flows on bridges accounting for the complexity of the hydrodynamic forces to which the bridge is subjected and the associated structural and functional response. Moreover, the impact of the reduced service on a bridge on the surrounding network is rarely addressed in the literature. Given this limited availability of models, this paper aims at establishing a multilevel modeling framework to address these issues in one combined approach. This aim is achieved by developing an integrated framework to assess the flooding impact on riverine bridges from the structural- to the network-level (Pregnolato et al., 2021b) and applying it to a real case study in the UK. This research tackles varying flow conditions (velocity and depth) to understand the structural response across given simulated flooding conditions. This work is novel since it represents a first attempt to couple CFD analysis with both Finite Element (FE) and network analysis for bridges subjected to flooding, in an effort to capture both the cause and effect of flooding. It is expected that this approach will be useful for understanding structural damage and functional loss for a range of bridges, and ultimately to assess risk for any coastal or riverine structure where large-scale water inundation is expected.

## 2 Method

This paper adopts a risk-based framework to assess the impact of high river flows to bridges and surrounding roads (Figure 1). The framework proposes a comprehensive method that encompasses the traditional four risk modules (hazard, exposure, vulnerability and consequences; Grossi and Kunreuther, 2005) and includes hydrodynamic force modelling, bridge susceptibility to the hazard, performance evaluation and network-level impact assessment. This study adopts specific models/software, but the precise chosen sub-models are not critical. In fact, all models/software are interchangeable, and it is reasonable to expect that the presented approach would be appropriate for software packages that ensure similar configuration.

The first step is to determine the intensity measures of flooding in terms of flow depth and velocity (see Section 2.1). For modelling fluvial flooding, most 2D hydrodynamic models can simulate flood depths and flow velocity, e.g. *LISFLOOD-FP* (https://bit.ly/3lstd4j) or *TELEMAC* (http://www.opentelemac.org/). Bridge information, such as geometry and design, can be retrieved through publicly available databases (e.g. the US National Bridge Inventory) or by coordination with local infrastructure managers and authorities; such information includes (but is not limited to) bridge dimensions, number of piers, material, design principle, foundation type. Unsurprisingly, the availability and accuracy of data vary from bridge to bridge and can influence the modelling outputs.

The second step consists of modelling the interaction between the water and the bridge, as well as the subsequent flood-induced loads. A simplified vulnerability and criticality assessment (Johnson and Whittington, 2011) include the evaluation of the local flow conditions and corresponding hydrodynamic forces that represent the load on the bridge structure using Computational Fluid Dynamics (CFD) techniques. Here, the C++ toolbox *OpenFOAM* is the adopted software, being open-source and particularly versatile for the development of customized numerical solvers (https://www.openfoam.org/).

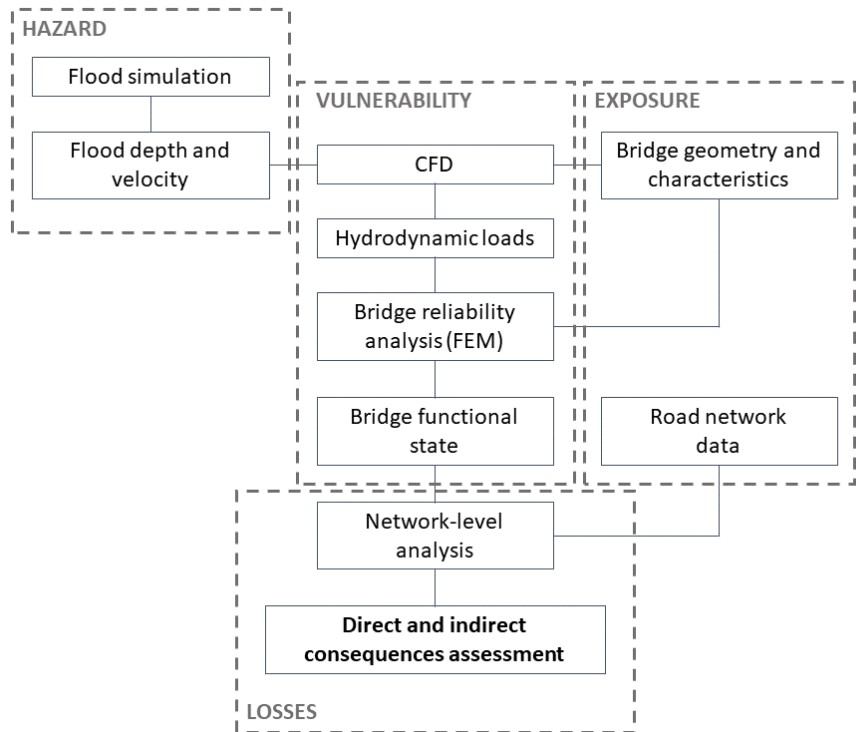

114

**Figure 1: The proposed risk-based methodological flowchart to integrate modelling of hydrodynamic forces, performance and network-level analysis. Acronyms: CFD - Computational Fluid Dynamics; FEM – Finite Element Model.**

The third step is to determine the response of the bridge subjected to flood through an advanced structural analysis approach such as Finite Element (FE) analysis. There are many available FE models, such as Abaqus FEA (www.3ds.com), ANSYS (https://www.ansys.com/en-gb), SAP2000 (https://www.csiamerica.com/products/sap2000) or the *OpenSees* software framework (McKenna et al., 2010). Mondoro and Frangopol (2018) described salient limit states for bridges subjected to hydraulic loads, and the subset studied in this paper (shown in Figure 2) includes yielding of the girders or piers, unseating or uplift of the girders, failure of the bearings, and excessive global displacement of the superstructure at which transient fluid-structure interaction is important (i.e., the CFD modeling approach is limited).

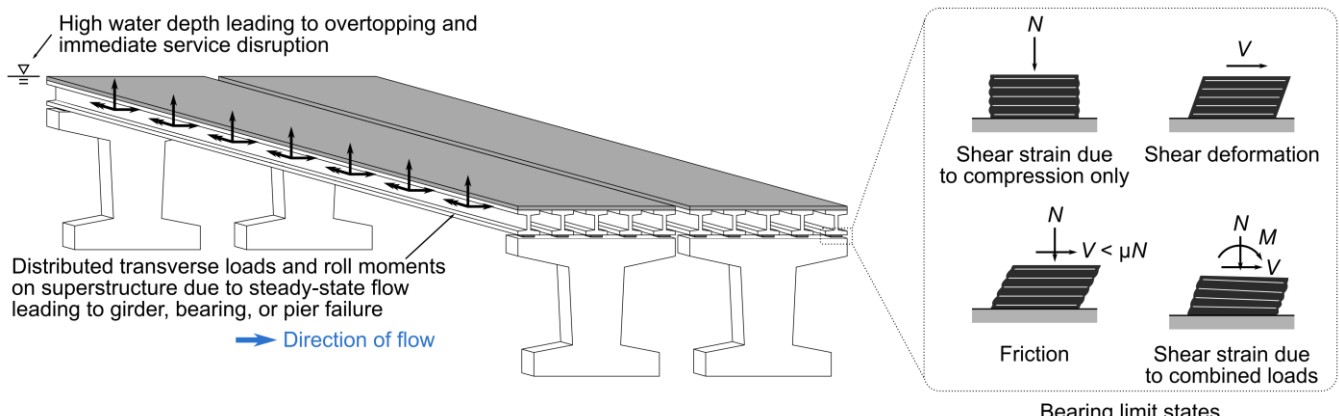

124

**Figure 2: Bridge failure states investigated due to flood loading.**

The general limit-states philosophy considers that specifications should satisfy "specified limit states to achieve the objectives of constructability, safety and serviceability" (AASHTO, 2017). In this work, the failure of a bridge is seen as twofold: *(i)* structural (also strength limit state), when the bridge deck, piers or foundation reach the ultimate limit state or permanent deformations; *(ii)* functional (also service limit state), when the bridge cannot perform its service as usual. A structural failure directly leads to a functional failure, e.g. the bridge collapses; preventive closure could also take place when bridge conditions are considered unsafe. Nevertheless, a bridge could be unserviceable but still structurally sound, e.g.

when floodwater or debris cover the deck. Hydraulic pressures (drag, lift and overturning moment) are assessed for
potentially dislodging the deck from piers, when submerged or partially sub-merged, and overtopping of the deck is
evaluated qualitatively from the CFD model. Though these limit states have significantly different long-term consequences,
both result in potential functional failure. The importance of long-term effects should be defined based on local
transportation needs.
The last step is to assess consequences, by including the impact of the bridge failure on the wider transport network.
Transport models such as *ESRI™ ArcGIS Network Analyst* (https://bit.ly/2GPMknl), *SUMO* (http://sumo.sourceforge.net/) or
*MatSIM* (https://www.matsim.org/) are suitable for computing routing and delays associated with a disrupted network link
(such as a closed bridge). Road network data are publicly available from sources such as Digimap®
(https://digimap.edina.ac.uk/), which provides Ordnance Survey road maps. These contain topographic information of roads
including name, location, length, capacity and type. After configuring the transportation network model with the collected
data, routing and accessibility can be investigated using network-based spatial analysis and transport appraisal techniques
(Arrighi et al., 2020; Pregnolato et al., 2016). This impact analysis links the structural damage of a bridge due to flooding
with the reduced performance of the local road network the bridge serves for, approximating the wider consequences.

### 2.1 Fluvial flooding simulation

Ideally, boundary conditions should be provided by gauging stations; however, no river gauges are present near the bridge of
interest, as is often the case in practical scenarios. This study adopted the 2D hydrodynamic model *LISFLOOD-FP,* which
allows to simulate flood depths and flow velocity to set up CFD boundary conditions for a flood scenario and from available
gauge data.
*LISFLOOD-FP* is a two-dimensional, spatially distributed, grid-based hydrodynamic model for simulating channel and
floodplain flows (Neal et al., 2009). The model dynamically simulates flood propagation in each grid cell at each time step,
on the basis of the local inertial formulation of the shallow water equations and an explicit finite difference method.
Numerically, this process involves calculating the momentum equation (the flow between cells given the mass in each cell)
and the continuity equation (the change in mass in each cell given the flows between cells) (Neal et al., 2018). The equations
underpinning the model, including their derivation, can be found in Bates et al. (2010) and de Almeida et al. (2012).
As input data, *LISFLOOD-FP* requires a DEM (Digital Elevation Model) of the area, channel and boundary condition
information (e.g. channel friction, width and depth, hydrograph and evaporation). Flow depth and velocity (for each cell) are
the output considered, since they represent the intensity measures of the hazard adopted by this study. The impact of bridges
on flow is not explicitly represented in this particular application.

### 2.2 Computational fluid dynamics (CFD) analysis

Three-dimensional (3D) CFD software is capable of resolving fine details of flood flow around bridges on a local scale such
as splashes, eddies, or flow separation, which cannot be captured by depth-averaged methods (such as *LISFLOOD-LP*).
Also, bridges present a problem for depth-averaged tools since the computational mesh is two-dimensional and cannot be
discretized vertically, which does not allow for a gap underneath a bridge superstructure. To accurately model such
behaviors is crucial when estimating flow-induced force demands, which requires the use of a fine, three-dimensional mesh.
Additionally, using higher fidelity, three-dimensional models allow for localized loads to be measured on individual faces of
a structure, which may be used to determine whether or not individual components fail versus entire structures (Winter et al.,

169 2017).

For this study, the three-dimensional CFD code *OpenFOAM* was selected. Flood flows were modelled using the interFoam
solver, which is a two-phase solver that relies upon Volume of Fluid (VoF) method (Tryggvason et al., 2011) to track the
interface between water and air phases. The underlying governing equations that are implemented in interFoam are the
Reynolds-averaged Navier-Stokes (RANS) equations, which are solved using a predictor-corrector or projection type of
method to solve for velocity and pressure fields, and advection equations for the volume fraction introduced by the VoF
method. More specifically, pressure-velocity coupling was achieved using the PIMPLE algorithm, which is a combination of
the Pressure-Implicit Split-Operator (PISO) and Semi-Implicit Method For Pressure-Linked Equations (SIMPLE). Since the
RANS system of equations does not constitute a well-posed system due to the so-called Reynolds stress tensor that arises
from the Reynolds-averaging process, a suitable turbulence model that introduces additional equations must be chosen to
close the system. For this study, the k-ω Shear Stress Transport (SST) model was used due to its ability to handle severely
separated flows near sharp corners better than other similar models such as the Standard, Renormalization Group (RNG), or
realizable k-ε models.

## 182 2.3 Structural analysis

A structural analysis approach is functional for: *(i)* simulating relevant structural response mechanisms, which differ based
on bridge type, and *(ii)* characterizing loading derived from the associated CFD model. Finite element (FE) analysis is
commonly employed in structural engineering to simulate the response of bridges to natural hazards for the purpose of
design and performance evaluation. Modern reinforced concrete and steel bridge structures are commonly formed of girders,
cap beams, and pier walls or columns which can be modeled as assemblages of line and spring elements; this approach is
common in practice and can be implemented in a wide variety of structural analysis programs. To model nonlinear response,
which is especially important when considering extreme loads associated with natural hazards, line elements may employ
concentrated or distributed plasticity formulations that make use of nonlinear hinges or fiber sections. Rotational, shear,
and/or axial spring elements can be used to simulate the response of discrete components such as connections and bearings.
Alternatively, continuum finite-element analysis can be employed for members if complex local response of components
(e.g. local buckling and/or deformation) is of interest; this approach is significantly more computationally expensive,
however. Other approaches, such as the discrete-element method, may be well suited for masonry bridges.
In this work, modeling with line and spring elements is performed, so this approach will be discussed in greater detail. The
considered bridge consists of a girder superstructure supported on reinforced concrete piers. *OpenSees* (McKenna et al.
2010) was selected as the analysis software due to its robust nonlinear modeling and scripting capabilities. This latter
capability is beneficial for performance evaluation using a suite of input parameters (in this case, a parameter sweep
characterizing different flood conditions). Moreover, the software is open source and therefore suitable for adaptation in
envisioned future work to enhance interactivity with *OpenFOAM*.
Component response and demands based on the structural analysis can be used to assign a damage state for the bridge. Here,
the structural damage is evaluated as slight, moderate, extensive, or complete damage based on the FEMA Hazus manual
(FEMA, 2003). Each of these damage states is associated with level of functionality and repair effort. The qualitative
description of damage states and average repair cost per $m^2$ ($ft^2$) is available in literature for hurricanes (Padgett et al., 2008)
and earthquakes (Hazus manual - FEMA, 2003); Gehl and D'Ayala (2018) offered a qualitative damage scale of potential
damage state and failure modes for the bridge components, which could be associated with functionality losses and remedial
actions. Table 1 adapts such literature to riverine flooding using additional works and expert opinion: it lists four identified
damage states (from slight to complete), and associated average repair cost and days of closure due to remedial works
(Werner et al., 2008; Gardoni, 2018; Lam and Adey, 2016).
**Table 1. Bridge damage states (Gehl and D'Ayala, 2018) associated to average repair cost per $m^2$ (Padgett et al., 2008; FEMA,**
**2003) and average days of closure due to repair (Werner et al., 2008; Gardoni, 2018; Lam and Adey, 2016).**

| Damage state | Description | Average repair cost (£/m²) | Days of closure |
|---|---|---|---|
| Slight | Minor damages such as cracking (shear | £1.45/m² ($0.25/ ft²) | 0-5 |

| | | | |
|---|---|---|---|
| | keys, hinges, deck) and spalling (hinges, columns) that require no more than cosmetic repair. Negligible scour. Some water and/or debris on deck. Full service, likely speed reduction of travelling vehicles. | | |
| Moderate | Moderate experience of shear cracks and spalling that still leave columns structurally sound. Moderate scour and moderate movement of the abutments. Significant water and/or debris on deck. The bridge is partially serviceable (e.g. alternating circulation, reduced capacity and load), but safe to use by emergency vehicles. | £36.54/m$^2$ ($6.28/ ft$^2$) | 5-12 |
| Extensive | Degradation of columns without collapse, shear and cracking leading to structurally unsafety. Significant residual movement at connections or major settlement approach. Delamination failure of individual bearings. Extensive scour of abutments. The bridge is closed to traffic. | £308.66/m$^2$ ($53.05/ ft$^2$) | 13-49 |
| Complete | Collapse of columns or connection losing all bearing support. Imminent deck collapse. Unseating of girders. Scour leading to foundation failure. The bridge is unserviceable. | £1102.77/m$^2$ ($189.43/ft$^2$) | >50 |


### 2.4 Fluid-structure coupling

The relationship between the CFD and structural analysis is critical to the implementation of the proposed framework as
outlined in the vulnerability analysis block in Figure 1. Both analyses must adequately represent the bridge geometry, and
the CFD output and structural analysis input loading must be compatible. Here, the coupling approach between *OpenFOAM*
and *OpenSees* is discussed, but the methodology is applicable to other software. It is noted that *OpenSees* alone is seldom
used to model structural response to fluids because of the complexity of the fluid loading and the required coupling
mechanism between fluid and solid solvers. As such, the present work is among the first of its kind using *OpenSees*. Other
recent research has sought to implement coupling between these multi-physics models. For example, Stephens et al. (2017)
demonstrated how *OpenSees* can be "loosely coupled" (i.e., with no interaction between CFD and FE models) with
*OpenFOAM* to characterize structural response due to sequential earthquake and tsunami loading. A similar loosely coupled
scheme is used here, where:
i.    the bridge superstructure (deck and girders) is modeled as a rigid, 2D cross section with a unit length out of plane
225         and subjected to steady-state flow at different water depths and velocities in *OpenFOAM*;
ii.    the steady-state reactions (output from *OpenFOAM*) on the cross section are recorded; and
iii.    the gravity loads and the steady-state reactions from *OpenFOAM* are applied as distributed, external loads on girder
228         line elements in a 3D *OpenSees* model of the full bridge.

It is noted that the bridge superstructure is rigid in the computational fluid dynamics model (an important simplification to
facilitate the analysis) but not in the finite-element model.

**2.5 Impact assessment**

The impact of a bridge failure in terms of consequences ($C$) includes direct consequences ($C_{dir}$) and indirect consequences
($C_{ind}$), which relate the surrounding transport network (Argyroudis et al., 2019; Kim et al., 2018). The total costs $C$ is
computed as (Eq. 1):

$$C = C_{dir} + C_{ind} = C_{repair} + C_{cleaning} + C_{detour} + C_{delay} \qquad \text{Eq. 1}$$

where $C_{repair}$ is the cost associated with repair or replacement of the bridge, $C_{clean}$ is the cost associated with the debris
removal (due to flooding), $C_{detour}$ is the additional vehicle operating due to the detour and $C_{delay}$ is the cost associated with trip
delays of normal traffic. Indirect costs may also include a fee for closing the bridge that the bridge owner has to pay to
transport operators/agencies (e.g. for railways, highways).
Table 1 (Sec. 2.3) was functional to compute $C_{repair}$. Average days of closure due to repairs are obtained via discussion with
national operators and existing literature (Werner et al., 2008; Gardoni, 2018; Lam and Adey, 2016). Values for $C_{clean}$ can be
researched among historic data of bridge owners, e.g. records from bridge inspection reports. $C_{detour}$ and $C_{delay}$ depend on the
network, type of vehicle and traffic flow; this study is limited to consider private cars and HGVs (Heavy Goods Vehicles, i.e.
over-3.5-tonnes-gross vehicle weight, including both articulated and rigid body types), for the sake of a contained
demonstration. According to standard transport appraisal procedures (e.g. DfT, 2009), the parameters are computed with Eq.
2 and Eq. 3 respectively. Considering an origin $i$, a destination $j$ and a vehicle type $z$:

$$C_{detour} = \sum_i \sum_j \sum_z q_{i,j,z}\, l_{i,j,z} VOC_z \qquad \text{Eq. 2}$$

$$C_{delay} = \sum_i \sum_j \sum_z q_{i,j,z}\, d_{i,j,z} VTT_z \qquad \text{Eq. 3}$$


$q$ is the volume of traffic, l is the incurred additional length, $d$ is the incurred additional time (delay), *VOC* is the extra
Vehicle Operating Cost (including fuel, tear and wear) and *VTT* is the Value of Travel Time, i.e. the non-monetary costs
incurred along the journey as time spent on transport. The additional length and travel time due to the detour are computed
using *ESRI™ ArcGIS Network Analyst*, setting the origin and the destination of the trip in opposite sides of the river as
demonstration (Pregnolato et al., 2016).

**3 Application and results**

The city of Carlisle is a flood-prone city (area: 1,040 km²; 2018 population: 108,387) located in the Northwest of England
(UK) (Fig. 3). Three road bridges connect the two parts of the town over the river Eden from North to South (the A689, A7
and M6 bridges) and a fourth one from West to East (Warwick bridge). The 2D hydrodynamic model *LISFLOOD*-LP was
set up to simulate a 1-in-500-year flooding scenario (Fig. 3b) for a domain covering 14.75 km$^2$ of Carlisle, at 5 m of
resolution. This simulation provided flow velocity and inundation height data.

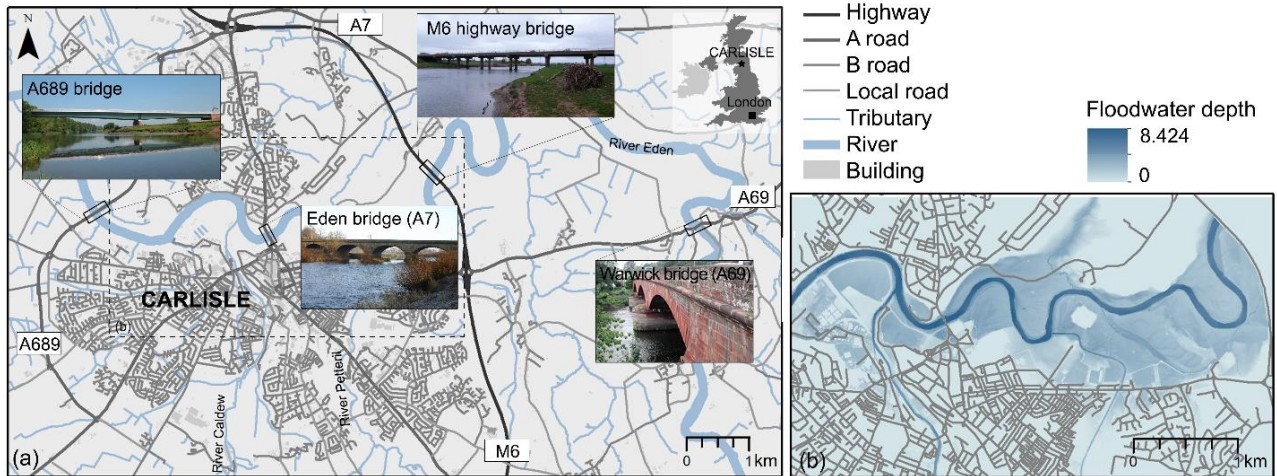

**Figure 3. The case study is the city of Carlisle, UK: (a) general overview of Carlisle upon the river Eden, connected North-South by three road bridges (the A689, A7 and M6 bridges) and West-East by the Warwick bridge (A69); (b) flood hazard map for Carlisle, as simulated with LISFLOOD-LP for a 1-in-500-year flood event.**

As a proof of concept, the M6 highway bridge over the River Eden was considered. The bridge is comprised of a girder
superstructure supported by hammerhead piers. A schematic model of this bridge is shown in Figure 4 with approximate pier
column (reinforced concrete), girder (concrete-encased steel), and bearing pad dimensions.

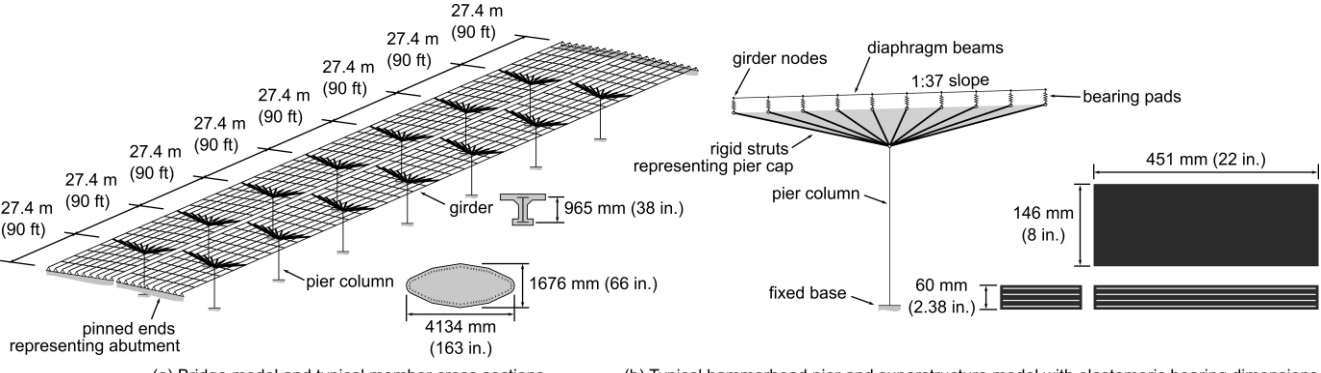

(a) Bridge model and typical member cross sections    (b) Typical hammerhead pier and superstructure model with elastomeric bearing dimensions
**Figure 4. Approximate geometry of M6 bridge as modeled in *OpenSees* including pier column, girder, and bearing dimensions**
**shown (not to scale).**
The pier columns are elliptically shaped and oriented to reduce hydraulic drag. The columns taper to a width of 4134 mm
and depth of 1676 mm at the base. The girders are supported on fixed, laminated elastomeric bearing pads with dowels at the
southern end of each span and free spherical bearings at the northern end. Salient bridge and flow input data are summarized
in Table 2.

**Table 2. Input data of this study for the exemplary CFD analysis of the M6 bridge (Carlisle, UK).**

| VARIABLE | DATA | SOURCE |
|---|---|---|
| Span length | 27.4 m | Drawings provided by Highways England |
| Superstructure width | 17.3 m | Drawings provided by Highways England |
| Superstructure weight (deck, girders, and diaphragm beams) | 514 kN/m | Derived from drawings |
| Flow Velocity | 1, 2, and 3 m/s | Modelled (LISFLOOD-LP) |
| Inundation Height | 12.5, 13.0, 13.5, 14.0, 14.5, 15.0, 16.0, 17.0, 18.0 m (from datum; +3.2 m) | Modelled (LISFLOOD-LP) |

## 3.1 CFD simulation and analysis

The CFD simulation was initiated at given inundation heightsand flow velocity, as modelled by the *LISFLOOD-LP* model
for a 1-in-a-500-year flood event at the site. The *OpenFOAM* model was set to simulate a range of flow velocity and depth
values above and below the calculated 500-year flood results in order to assess how varying the depth and velocity affected
the resulting bridge performance. Flow velocities and depths were extracted from LISFLOOD-LP in proximity of the bridge,
and also compared with historical data (e.g., the peak flow recorded at Sheepmount, UK in December 2015 was equal to
1680.0 $m^3$/s; EA, 2016) and inspection reports. The statistics for the velocity (both in its actual flood flow direction and also
normal to the bridge) were computed from the *LISFLOOD*-LP velocity vector (Vx, Vy) and maximum water depth data,
considering maximum values for both quantities over the whole flood simulation. The 500-year return period flood showed
velocity values up to roughly 3.5 m/s and max flood depth up to 17 m near the M6 Bridge. These statistics motivated using a
range of steady-state velocities of 1-3 m/s and inundation elevations of 12.5-18 m above datum in the *OpenFOAM*
simulations. The bridge superstructure was positioned such that the bottom of the bridge's lowest girders and the highest
point of the top of the bridge deck were at elevations of 12.375 m and 14.425 m respectively, relative to the datum, which
was 3.2 m below the riverbed's lowest point. Flow rates corresponding to the range of selected flow velocities and depths
were specified at the inlet boundary of the *OpenFOAM* model, using the variableHeightFlowrate boundary condition. To
model the free-surface flow of the Eden River interacting with the M6 bridge in *OpenFOAM*, the interFoam multiphase fluid
flow solver which utilizes the Volume of Fluid method for interface tracking was used along with the k-ω SST turbulence
model to resolve turbulent flow behaviors. Default *OpenFOAM* values for air-water physical fluid properties (densities: $\rho_{air}$ =
1 kg/$m^3$, $\rho_{water}$ = 1000 kg/$m^3$; kinematic viscosities: $\nu_{air}$ = 1.48(10$^{-5}$) $m^2$/s; $\nu_{water}$ = 1.0(10$^{-6}$) $m^2$/s; surface tension: σ = 0.07
N/m) and turbulence model coefficients were used for all simulations. A full summary of all *OpenFOAM* boundary
conditions is provided in Table 3.
**Table 3. *OpenFOAM* model boundary conditions.**

| Boundary | OpenFOAM Simulation Field Variables | | | | | | |
|---|---|---|---|---|---|---|---|
| | alpha | epsilon | k | nut | omega | p_rgh | U |
| InletWater | variableHeight-Flowate | fixedValue | fixedValue | calculated | fixedValue | zeroGradient | variableHeightFlow-RateInletVelocity |
| InletAir | inletOutlet | inletOutlet | inletOutlet | calculated | inletOutlet | totalPressure | pressureInletOutlet-Velocity |
| OutletWater | zeroGradient | zeroGradient | zeroGradient | calculated | zeroGradient | zeroGradient | inletOutlet |
| OutletAir | zeroGradient | zeroGradient | zeroGradient | calculated | zeroGradient | totalPressure | pressureInletOutlet-Velocity |
| Right | empty | empty | empty | empty | empty | empty | empty |
| Left | empty | empty | empty | empty | empty | empty | empty |
| Bottom | zeroGradient | epsilonWall-Function | kqRWall-Function | nutkWall-Function | omegaWall-Function | fixedFlux-Pressure | noSlip |
| Atmosphere | inletOutlet | inletOutlet | inletOutlet | calculated | inletOutlet | totalPressure | pressureInletOutlet-Velocity |
| Bridge | zeroGradient | epsilonWall-Function | kqRWall-Function | nutkWall-Function | omegaWall-Function | fixedFlux-Pressure | noSlip |


To reduce computation time and provide conservative results, a unit width segment of the bridge superstructure located
above the deepest point of the Eden River beneath the M6 Bridge was analyzed in *OpenFOAM*, which resulted in a 2D
simulation that drastically reduced the mesh cell count compared to a full 3D simulation of the entire bridge. Additionally,
the out-of-plane direction components of the flow were neglected in all simulations by using the empty type of *OpenFOAM*
boundary condition, ensuring the simulations were truly 2D. This setting allowed for more simulations to be run using a
wider range of flooding conditions in less time while conducting the parametric study. As shown in Fig. 5, the model
measured forces on 20 individual components along the cross-section of the bridge superstructure segment corresponding to
each girder and its tributary width of the bridge deck.

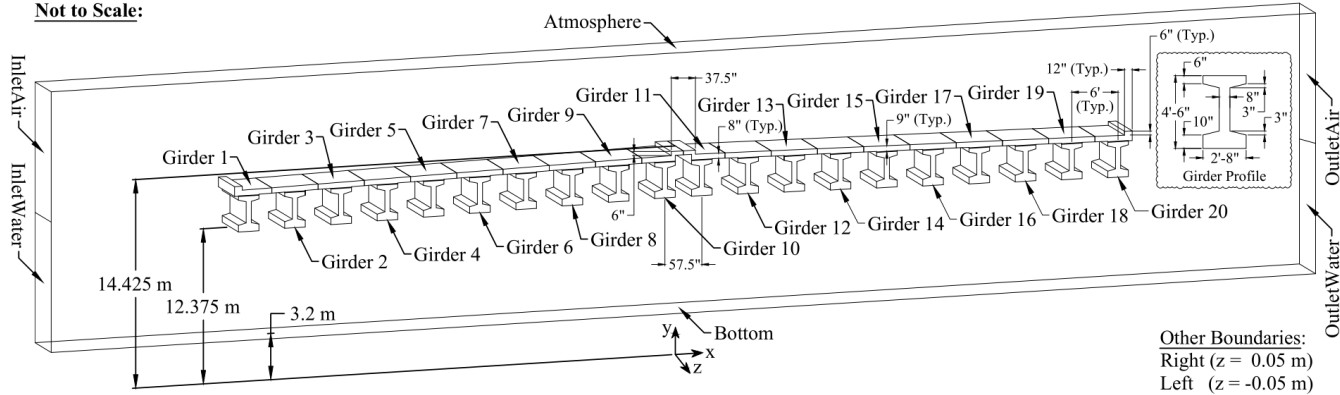

**Figure 5. OpenFOAM model geometry and boundary conditions**
Fig. 6 shows converged OpenFOAM fluid load outputs for each bridge component at all inundation levels, providing an
example for an initial flow velocity of 3 m/s, which corresponds to the worst-case scenario simulated in this study. Since the
simulations were 2D, load values are expressed in units of force or moment per meter of bridge deck width.
The horizontal forces presented in Fig. 6a show significant peaks at the bridge deck edges for components 1 and 20, due to
the asymmetric pressure distributions that these components experienced when comparing their upstream and downstream
faces than the interior components (which were shielded from higher velocity flows by the exterior components).
Additionally, the exterior components included the traffic barriers, which significantly increased their surface area on which
fluid pressure acted compared to the interior components.
At the upstream edge of the deck, component 1 absorbed the primary impact of the incoming flood flow at its peak velocity
since it was on the upstream side of the deck, resulting in it carrying the largest positive horizontal forces. At the
downstream edge of the deck, component 20 was initially subjected to positive horizontal forces due to the flow impacting
its bottom flange and the lower portion of its web, but for flow heights greater than 14.0 m, its horizontal force decreased
until it became negative by a flow height of 16.0 m. The gradual decrease in component 20's horizontal force may be
attributed to differences in the vertical surface areas of and the flow velocities near its up- and downstream sides that resulted
in larger fluid pressures acting on the downstream faces than the upstream faces. The total vertical surface area of the
downstream faces was larger than that of the upstream faces by an amount equivalent to the deck section, which provided
additional area on which fluid pressure acted in the upstream direction. Complex flow characteristics that contributed to the
velocity differences include: 1) the recirculatory flow patterns between the girders of components 19 and 20 and in the
corner between the deck top and traffic barrier that led to reduced pressures on upstream faces of component 20; 2) the
turbulent eddies that were shed off of the leading edge of component 20's girder bottom flange that redirected the flow
toward the downstream faces of component 20; and 3) the flow over the top of the bridge deck rejoining the flow beneath the
bridge at the downstream edge of the deck, which contributed to the formation of turbulent eddies in the bridge deck wake.
Also, in the event that any air was trapped between girders, a lesser water level between the girders would further decrease
component 20's horizontal force.
The vertical forces shown in Fig. 6b are of similar magnitude to the horizontal force values in Fig. 6a. When the flow height
was small prior to the flood overtopping the bridge (i.e. 12.5 m to 13.0 m), the vertical forces on both halves of the bridge
were roughly uniform except for the components nearest to the upstream edge of the bridge. In these cases, the vertical
forces on components 1 - 3 decreased due to fluid pressure acting downward on the top of the girder bottom flanges. For
flow heights of 13.5 m to 16.0 m, the vertical forces on the upstream half of the bridge initially increased due to buoyancy
forces increasing (due to increasing flow depth), but it started to decrease at a flow height of 14.0 m as the flood began to
overtop the bridge. By a flow height of 17.0 m, the bridge was submerged enough that buoyancy caused the vertical forces
on the upstream half of the bridge to increase again.
For the downstream half of the bridge, uplift due to buoyancy increased until a flow height of 15.0 m. At this point, the flow
overtopped the superstructure crest at the midpoint of the bridge. This change in flow behaviour caused the vertical forces on
the downstream half of the bridge to decrease until the bridge was sufficiently submerged at a flow height of 18.0 m.
Overturning moment results acting about the $z$-axis of each bridge component are shown in Fig. 6c. Similar to the horizontal
forces shown in Fig. 6a, the extreme overturning moment values occurred at the edge components of the bridge deck,
whereas the interior components experienced much smaller overturning moments since they are shielded by the edge
components. At flow depths less than 14.0 m prior to the flood overtopping the barriers, increasing positive moment values
for the component 1 at the upstream edge indicate that a counter-clockwise rotation would occur, which would cause the
upstream side of the bridge component to move downwards, whereas downstream side would move upwards. This process is
due to the lesser depth flows only impacting the bottom flanges and lower parts of the webs of the girders. As the flow depth
increased, the position of the resultant horizontal force gradually increased until it moved above the centroid of the
component 1 girder, about which moments were summed. This effect resulted in a trend reversal such that the moment
decreased with increasing flood depth until for depths of 16.0 m or more clockwise rotations of component1 occurred due to
flow overtopping and eventual full submersion the bridge. At the opposite deck edge, component 20 experienced a similar
trend switch, where its overturning moment initially decreased for flow depths less than 14.0 m, but increased for 14.0 m or
more, changing from counter-clockwise to clockwise rotation beginning with the 16.0 m flow height case.

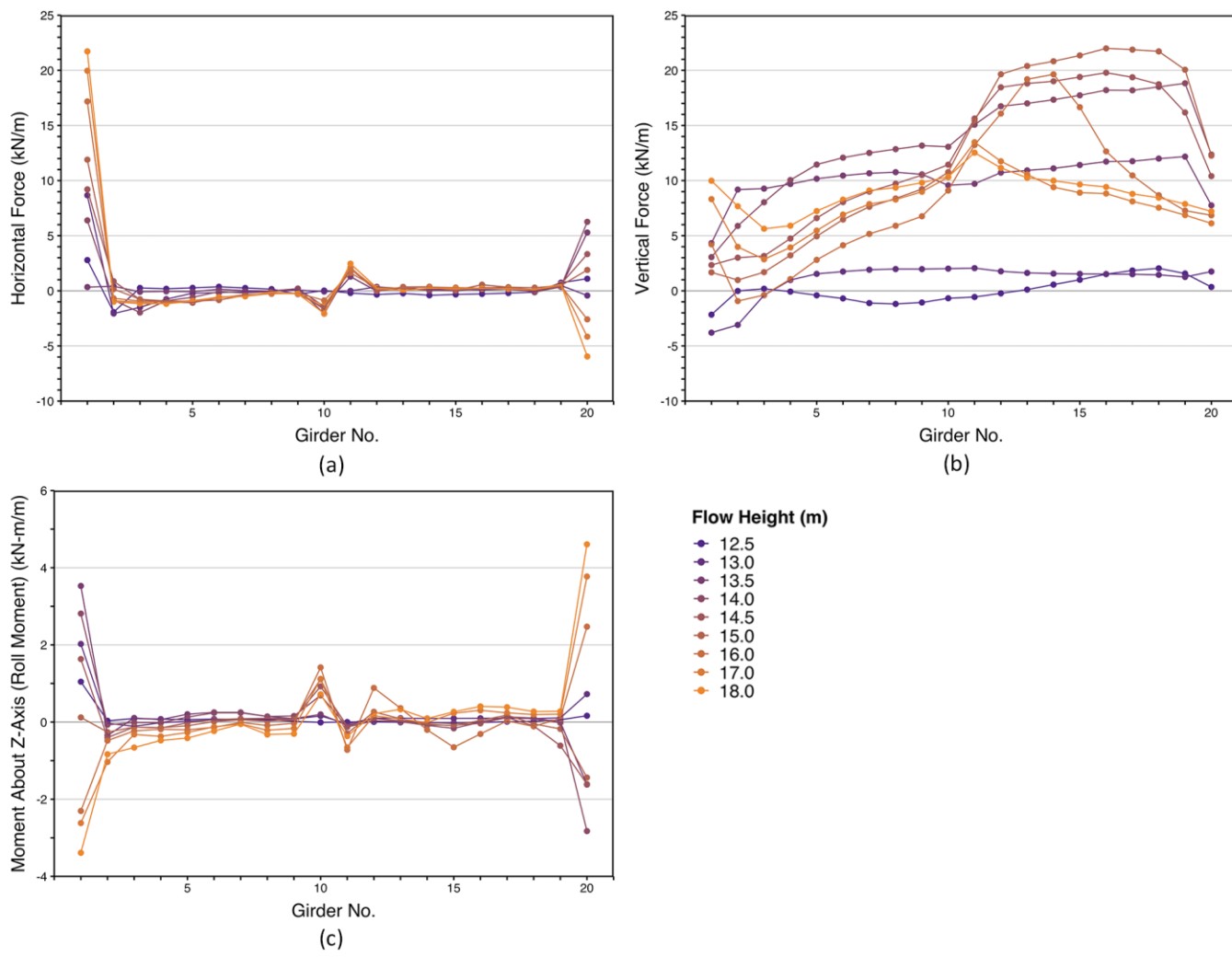

Figure 6. Converged simulated component loads for flow velocity equals 3 meters/second per girder component; (a) shows horizontal (x-direction) loads applied in kN per meter bridge width; (b) shows vertical (y-direction) loads applied in kN per meter bridge width; (c) shows moment about the z-axis (i.e. roll moment) in kN-m per meter bridge width.

**3.2 Structural analysis and damage assessment**

The *OpenSees* model was developed using fiber-based line elements for the reinforced-concrete pier columns and preflex girders (a form of prestressed, concrete-encased steel beams). Nonlinear concrete (Concrete02) and steel (Steel02) constitutive models were employed to simulate uniaxial material response in the fibers. All concrete was assumed to have a compressive strength of 34.5 MPa. The steel reinforcement and encased structural steel was assumed to have yield stresses of 276 MPa and 379 MPa, respectively. The girders ends were connected to pier caps (modeled as rigid) via linear-elastic springs to represent bearings. The free spherical bearings were modeled as roller boundary conditions. The steel-laminated elastomeric bearing pads were modeled with lateral, vertical, rotational, and torsional stiffnesses based on linear theory of bearings as described by Stanton et al. (2008). The elastomeric bearing dimensions are shown in Figure 4; each had two, 13-mm-thick layers of elastomer reinforced with 3-mm steel plates. The elastomer was assumed to have a bulk modulus of 3100 MPa and a shear modulus of 0.76 MPa; the bearing dimensions and material properties led to the stiffness parameters defined in Table 4. The bearing spring elements were connected to rigid links which simulated pier cap beams, providing a load path between the girders and pier columns. The bridge abutments were founded on rock on the north side and piles on the south side; both abutments were modeled as rigid. The piers were founded on rock and pier columns were modeled as fixed. It is noted that many bridge foundations are vulnerable to scour, especially under flood conditions; however, the piers and abutments of the considered bridge are founded on rock, thus scour is not a concern for this structure (and in general scour and soil-structure interaction effects are beyond the scope of the present work).

**Table 4. OpenSees elastomeric bearing spring stiffnesses**

| Stiffness type | Direction | Value |
|---|---|---|
| Axial | — | 142 kN/mm |
| Shear | — | 1.69 kN/mm |
| Rotational | Deformation in short-axis direction | 311 kN-m/rad |
| | Deformation in long-axis direction | 2350 kN-m/rad |
| Torsional | — | 17.9 kN-m/rad |


To analyze the bridge, gravity loads were first applied based on the self-weight of the structural components; no live loads
were considered. The lateral forces, vertical forces, and roll moments determined from *OpenFOAM* were then applied as
distributed loads in *OpenSees* on each bridge girder (i.e., over all eight spans with 20 girders per span); this step is the key
link between the CFD and structural models.
Under the range of loading investigated, yielding or cracking was not detected in the girders or columns, and the simulated
hydraulic forces were not large enough to overcome the self-weight of the structure, which would result in uplift of the
superstructure. However, the elastomeric bearing pads sustained large shear demands near the design limits specified by
Section 14.7.5 of the AASHTO *LRFD Bridge Specification* (2017). Specifically, the elastomeric bearings were evaluated
for:
i.    loss of frictional resistance between the bearing and girder based on the ratio of shear and normal forces on the

391         bearings,

ii.    excessive shear deformation, and
iii.    excessive shear strain due to combined axial load, rotation, and shear deformation.
The solid lines in Figure 7 compare maximum shear forces, deformations, and strains in any of the elastomeric bearings for
each of the loading scenarios investigated; Figures 7a, 7c, and 7e show these engineering demand parameters versus flow
velocity and Figures 7b, 7d, and 7f show corresponding values with respect to flow height. The data suggest that peak
steady-state demands on any of the elastomeric bearings in the bridge occur around a flow height 15 m, at which point the
bridge has just reached full inundation. In addition, below a flow height of 15 m, demands consistently increase with
velocity; such increases in demand after full inundation are not consistently observed, which suggests that the loading is
primarily associated with hydrodynamic effects that are a function of the effective area of the cross-section, and may also be
affected by the fact that the flow around the superstructure is less turbulent. To expand the data set, linear extrapolation to
flow velocities of up to 6 m/s are shown in Figures 7a, 7c, and 7e as dotted lines with open markers. It is noted that the plots
in Figure 7 show peak demands across all elastomeric bearings in the bridge, and the actual extent of damage depends on the
progression of failure in multiple bearings.
The Commentary to the AASHTO *LRFD Bridge Specification* (2017) states a coefficient of friction of 0.2 between
elastomeric bearings and concrete is appropriate for design, and this limit is used here to evaluate potential girder unseating
due to loss of frictional resistance. For the purpose of this evaluation, dowel resistance is neglected, though this effect could
prevent unseating in practice. Figures 7a and 7b plot the peak ratios of shear-to-normal forces across all bearings on the
bridge, and it can be observed that the bearings are well below the limit suggested in the AASHTO Commentary (which is
labeled as $\mu_{max}$ and shown as the grey line). However, it must be noted that the coefficient of friction may be lower than
expected under wet conditions and that the lateral hydrodynamic loading can be significant, increasing vulnerability of
unseating due to debris impact. To illustrate how the sequential fluid-structure modeling results may be applied, a highly
conservative, reduced coefficient friction of 0.1 is considered. Using this threshold, the results indicate flow conditions for
which the given frictional resistance is approached or exceeded: 13.5-m flow depth with velocity of at least 6 m/s, 15-m flow
depth with velocity of at least 5 m/s, 18-m flow depth with velocity of at least 6 m/s.
Figures 7c and 7d show peak shear strains due to loading perpendicular to the short edge of the bearing pad (see Figure 4b)
due to combined axial load ($\gamma_a$), rotation ($\gamma_r$), and shear ($\gamma_s$). The shear strains are computed based on Eqs. 4-6 based on the
AASHTO *LRFD Bridge Specification* (2017).

$$\gamma_a = D_a \frac{\sigma_s}{GS_i} \qquad \text{Eq. 4}$$

$$\gamma_r = D_r \left(\frac{L}{h_{ri}}\right)^2 \frac{\theta_s}{n} \qquad \text{Eq. 5}$$

$$\gamma_s = \frac{\Delta_s}{h_{rt}} \qquad \text{Eq. 6}$$

In the above equations, $D_a$ and $D_r$ are empirical coefficients, $\sigma_s$ is the average compressive stress, $G$ is the shear modulus, $S_i$
is the shape factor of the $i^{\text{th}}$ internal layer, $L$ is the bearing length perpendicular to the axis of rotation, $h_{ri}$ is the thickness of
the $i^{\text{th}}$ internal elastomeric layer, $h_{rt}$ is the total thickness of the elastomer, $\theta_s$ is the rotation demand, $n$ is the number of
interior elastomeric layers, and $\Delta_s$ is the shear deformation. Note that $\sigma_s$, $\theta_s$, and $\Delta_s$ are outputs from the structural analysis;
the rotation demand, $\theta_s$, includes 0.005 rad of rotation due to misalignment. For design per the AASHTO *LRFD Bridge*
*Specification* (2017), the combined shear strain due to these actions should not exceed 5.0, and this criterion is satisfied in
the analyses (all values, including extrapolated values, are below the grey line in Figures 7c and 7d).
The shear deformation demand on the bearing $\Delta_s$ is shown to be more critical than the combined shear strains: Figures 7e
and 7f show these data with the annotated shear strain limit of $h_{rt}/2$ in grey; this limit is also based on the AASHTO
*Specification* (2017). The demand is clearly largest for a flow height of 15 m, and it increases linearly with the flow velocity.

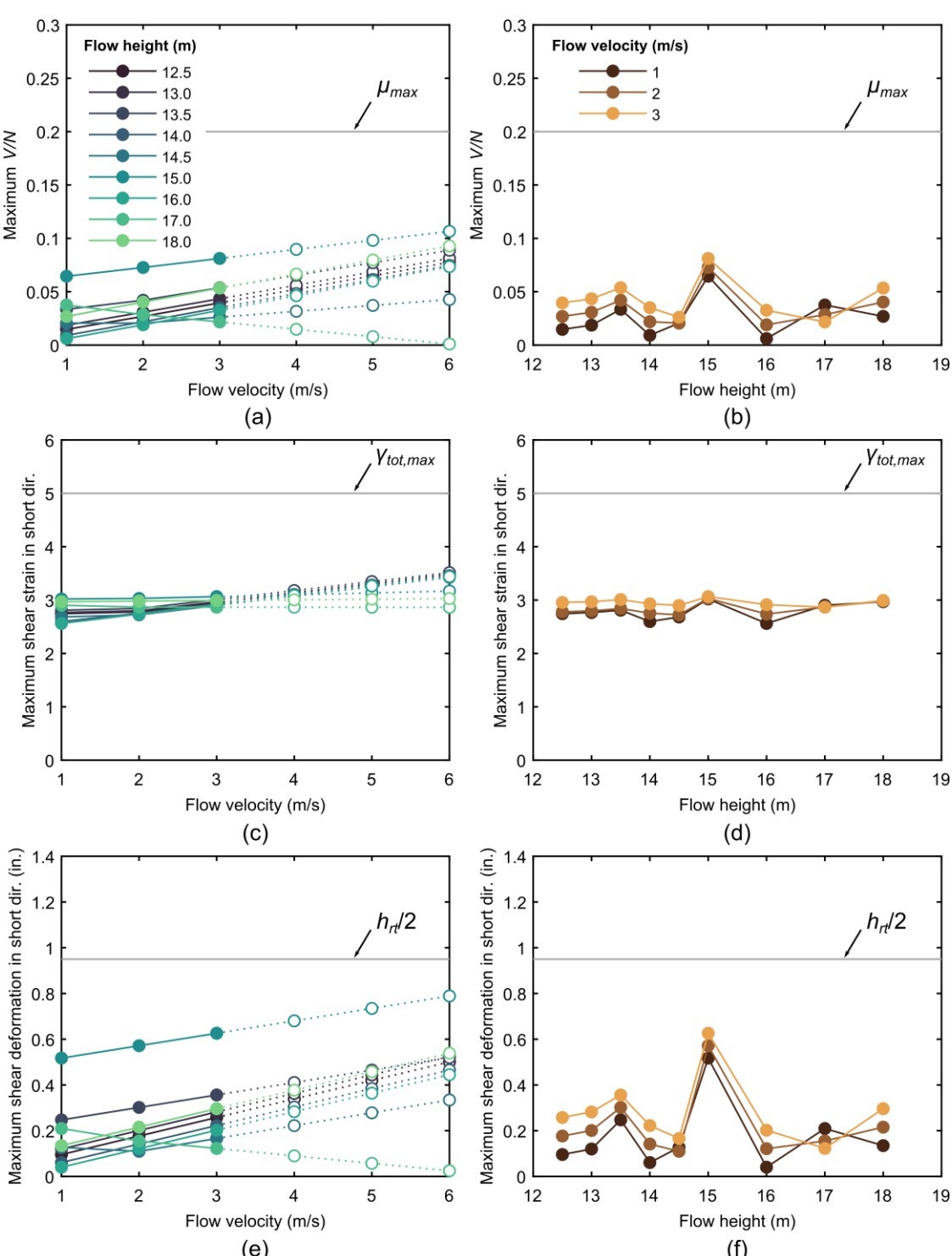


**Figure 7.** Maximum simulated demand on elastomeric bearings in M6 bridge, including (a)/(b) shear force, (c)/(d) total shear strain due to combined axial, moment, and shear demands, and (e)/(f) shear deformation; plots on left show demand versus flow velocity and plots on right show demand versus flow height.

## 3.3 Network impact and consequence assessment

The results of the loosely coupled CFD and structural analyses described in Sec. 3.2 suggest a potential for either girder unseating due to loss of frictional resistance or excessive shear deformation, which may lead to debonding and delamination for this particular bridge. In addition, damage associated with these limit states is most expected at a flow height of 15 m and flow velocity of at least 5 m/s. The impact of damage in this flood scenario is therefore considered in this section. Based on Table 1, the damage state is estimated as moderate because: *(i)* the bearings approach but do not exceed limit states, *(ii)* scour is assumed to be insignificant compared to damage to the superstructure and bearings, and *(iii)* water overtops the bridge deck. A moderate damage state implies the bridge closure for 5-12 days (see Table 1). In the case of the M6 bridge,

its closure causes disruptions to all southbound and northbound users that are travelling along the M6 (Figure 8). Compared
to the baseline journey, results show that private cars are delayed by 12 minutes and have additional ca. 9 km due to
rerouting. HGVs cannot travel via the historic Eden Bridge (city center) and are subjected to a longer rerouting, which leads
to 19 minutes and ca. 20 km of delay and additional travelling respectively.

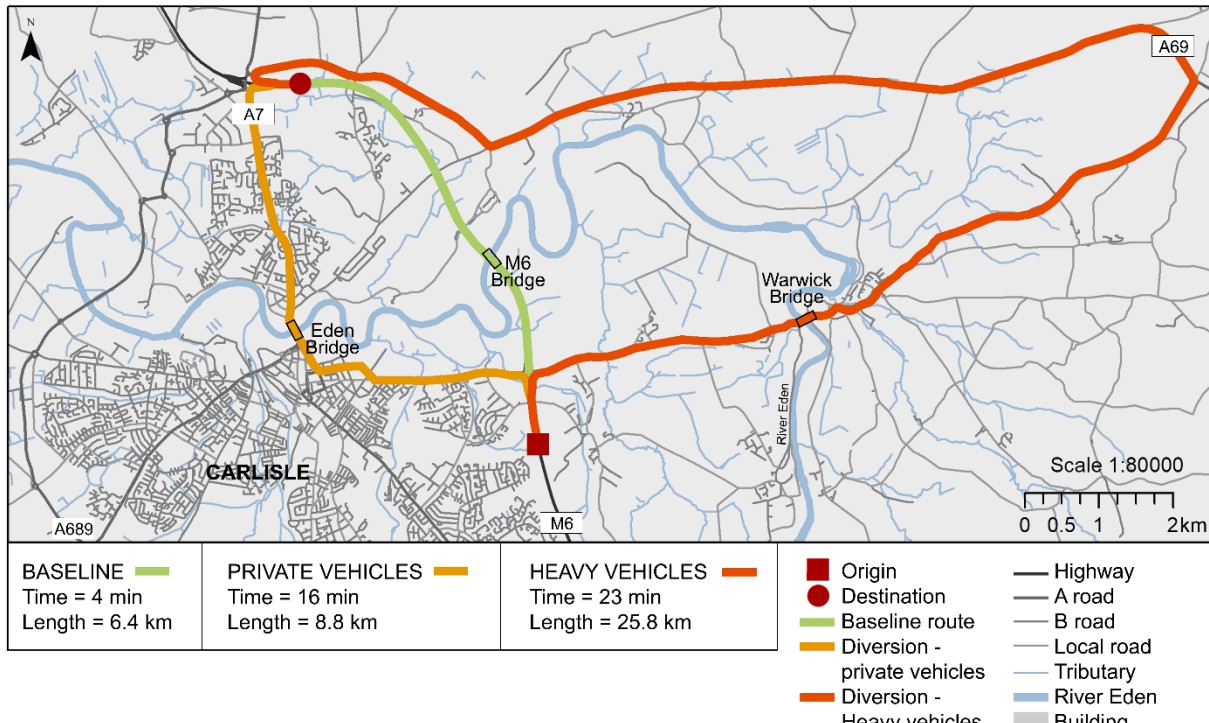


**Figure 8. Routes for crossing the river Eden along the highway in baseline and disrupted conditions; private and heavy vehicles**
**are rerouted on different journeys when the M6 bridge is disrupted.**
The cost of the impact due to the M6 bridge disruption is computed in terms of direct and indirect consequences using Eq. 1;
output and input values are specified in Table 5.
**Table 5. Output and input data for the impact cost calculation considering disruption due to an extreme flood event on the M6**
**bridge in Carlisle. Acronyms: VTT – Value of Travel Time; HGV - Heavy Good Vehicle; VOC – Vehicle Operating Cost; ADT -**
**Average Daily Traffic.**

| | VARIABLE | DATA | SOURCE |
|---|---|---|---|
| INPUT | Average repair cost (£/m$^2$) | £36.54/m$^2$ | Table 1 |
| | Time for repairs ($T_{repair}$) | 7 days | Table 1 |
| | VTT for HGVs | £10.10/hour | DfT (2009) |
| | Delay for HGVs | 19 min | computed |
| | Detour length for HGVs | 19.4km | computed |
| | VOC for HGVs | 37.668 p/km | Blakemore (2018) |
| | ADT for HGVs | 1833 veh/day | UK national statistics |
| | VTT for average private vehicles | £6.81/hour | DfT (2009) |
| | Delay for average private vehicles | 12 min | computed |
| | Detour length for private vehicles | 2.4 km | computed |
| | VOC for private vehicles | 25.47p/km | Yurday (2020) |
| | ADT for average private vehicles | 28602 veh/day | UK national statistics |
| OUTPUT | $C_{repair}$ | £7,308.00 | computed |
| | $C_{clean}$ | £29,476.00 | Panici et al. (2020) |

| | | |
|---|---|---|
| $C_{detour}$ | £30,878.65/day | computed |
| $C_{delay}$ | £44,818.47/day | computed |
| TOTAL | £566,663.81 | |


The values of Value of Travel Time (VTT) of HGVs (Heavy Good Vehicles, working condition) and average private cars
(unspecified conditions) can be found in the UK Department for Transport (DfT) appraisal methods, illustrated in the Cost
Benefit Analysis (COBA) manual (DfT, 2009). Data regarding the additional travel time for rerouting has been computing
via transport model (Sec. 2.5) and verified with Google Maps (Figure 8); for the UK, topological road network links are
freely available nationwide. Data regarding Average Daily Traffic (ADT) flow are freely available
(http://webtris.highwaysengland.co.uk/) and were obtained by considering the annual northbound and southbound flows for
the relevant sites (36,670 veh/day: Site 9538/2 on link M6 southbound and Site 9540/2 on link M6 northbound; 2019 data),
considering the traffic composition at 78% for private cars and 5% for HGVs (DfT, 2019).
The repair cost ($C_{repair}$) was computed using Table 1 and assuming 7 days (average) of bridge closure; the cost of debris
removal was obtained by looking at the highest cost for a single event in the UK (Panici et al., 2020), since the simulated
flooding is an extreme and rare event. The additional vehicle operating due to the detour per day ($C_{detour}$) was calculated
using Eq. 2; the cost associated with trip delays ($C_{delay}$) was calculated using Eq. 3.
For the case study undertaken (Carlisle, UK; 1-in-a-500-ys event), the total cost of the flood impact to the bridge is
£566,663.81, considering seven days of bridge closure. The largest proportion (93.5%) of this cost is due to the indirect cost
of rerouting traffic (£75,697.12 per day of closure, i.e. £529,879.81); the 6.5% of the total cost is due to direct damages only
(£36,784.00).
**4 DISCUSSION AND FUTURE RESEARCH**
This study developed an integrated method that uses a multiphysics, multilevel approach for assessing the effect of flooding
hazards on a local transportation network. For the city of Carlisle (UK), a 1-in-500-years flooding event was simulated and
the resulting hydrodynamic forces on the highway bridge (M6) modelled. While simulated hydrodynamic forces and Finite
Element (FE) analysis did not show uplift failure, overtopping of the bridge is shown to occur at inundation heights of 14 m
and above. Given the potential for flood-related disruption of traffic, overtopping should be considered temporary network
failure in its own right. The elastomeric bearings supporting the bridge girders approached shear deformations near design
limits at a flow height of 15 m, and a potential loss of frictional resistance between the elastomer and concrete is also
observed. While these limit states were not exceeded for flow velocities up to 3 m/s, extrapolation to faster flow rates
suggests higher potential for damage. Under this hypothesis, the bridge would lose immediate functionality at a flow height
of between 13.5 and 14.0 m due to inundation of the deck even if the structure sustains no damage. The impact analysis
showed that indirect damages covered the 93.5% of the total cost of damages to the bridge, proving that limiting the
assessment to repairs and debris cleaning would greatly underestimate the impact of flooding to bridges.
The produced outputs are conceptual results and thus approximate and indicative for multiple reasons. First, there is a dearth
of UK-specific data regarding bridge repairs, duration time of repair, etc.; research or survey to solicit post-flood data are
highly recommended to improve impact estimates. For example, a bridge could be partially closed during repairs (according
to its damage state) and allow traffic in one direction. Second, the modeling approach presented herein used several
intentional simplifications for demonstration purposes, including reducing the CFD domain, neglecting soil-foundation
effects and scour modeling, and assumed rigidity of the structural system among others. In scenarios where these issues (or
others) may be of more concern for a particular bridge, the fidelity of the modeling approach could be improved.
Additionally, the failure states presented here may not translate broadly to the general bridge inventory, but additional or

alternative structural/functional failure states could be applied. Third, the impact analysis was limited to private cars and HGVs for demonstration purposed; however, advanced transport appraisal could better capture users' choices and the engineering response of lifelines by including a wider range of vehicles categories and traffic scenarios. In terms of impact, the presence of floodwater on the roads is not simulated for limiting the focus of this work on riverine flooding and the bridge impact consequences; for properly analyzing the flooding impact to road networks, simulation of surface water flooding should be undertaken; this analysis would be a study on its own, and currently out of the scope of this piece of research. Flood impact on other parts of the network would limit the capacity of the alternative routes, causing additional delays to the traffic; thus, obtained results represent an underestimation of the overall systemic cost. Nevertheless, the proposed approach of impact analysis can give modelers and analysts a comprehensive method for assessing susceptibility to flooding and relative consequences at systemic level and the case study presented here represents an archetype for this approach.

Thus, the importance of this study consists in the proof of concept of a new holistic methodology which uses a multilevel approach to improve the fidelity of network failure predictions, taking advantage of seemingly disparate physical models. The computed hydrodynamic forces were applied directly into a traditional FE model to predict the global structural response to identify critical structural components and damage states. Notably, the hydrodynamic forces induced large demands on bearings that are often not considered in design. Because of the critical nature of bridges to a transportation network, the impact analysis revealed that indirect cost cover almost all the total cost due to flooding; this consideration is fundamental for infrastructure owners and managers when managing assets and budgets.

Next steps of this study will analyze the impact of the closure for a second bridge (e.g. the masonry arch Eden Bridge – data permitting), in isolation first and then in combination with the M6 bridge. Future work should investigate the impacts of other limit states which could result in total or partial bridge closure; a wider range of bridge types should be investigated too. Such analyses would benefit from 3D CFD and FE models to help refining demands on the structure and reducing uncertainty in the predicted bridge performance. Ultimately, this approach can be applied to any coastal or riverine structure where large-scale water inundation is expected.

**5 CONCLUSION**

This study focused on riverine bridges prone to failures during flood events. This study established rigorous practices of Computational Fluid Dynamics (CFD) for modelling hydrodynamic forces on inundated bridges, and understanding the consequences of such impact on the surrounding network. The hydrodynamic forces were modelled as demand on the bridge structure and inputted into a vulnerability analysis of the structure; the performance evaluation s showed a moderate damage state of the bridge which was used to approximate the overall direct and indirect consequences. For the city of Carlisle (UK) and a 1-in-500-years flooding, results showed that the flood impact to the M6 bridge (highway bridge) caused more than £500k of damages of which 93.5% indirect damages (rerouting and delays). The relevance of this work resides in the integrated method that couple practices of CFD with performance and network analysis, which allows to estimate the cost due to flooding impact to a bridge considering the surrounding transport system. Infrastructure owners and managers, as well as modelers and researchers, should build on this work to better predict local fluid pressures that may lead to bridge structural failure and related network-wide consequences.

**DATA AVAILABILITY STATEMENT**

All relevant and publicly available data will be shared via the DataBris repository of the University of Bristol if the paper will be accepted for publication; data sources are clearly specified throughout the paper.

**ACKNOWLEDGEMENTS**
MP was supported by the Engineering and Physical Sciences Research Council (ESPRC) LWEC (Living With
Environmental Change) Fellowship (EP/R00742X/1 and 2). The authors also grateful acknowledge: Mark Pooley at
Highways England; John L. Kelsall at Phoenix Architecture & Planning; Mohammad Fereshtehpour at Ferdowsi University
of Mashhad.

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
