# Peer review of "Assessing flooding impact to riverine bridges: an integrated analysis"

_Natural Hazards and Earth System Sciences, 2020_

## Referee Comment (RC1) · Anonymous Referee #1 · 9 Feb 2021

The manuscript presents an integrated framework for the assessment of flood impact on riverine bridges and the road network they connect, with the merit of including all the most relevant aspects of the problem, from the hydraulic to the structural and road network ones. The methodology is demonstrated on a UK case study, and it is intended to be applicable also elsewhere. Most of the various steps in the methodology are based on approaches that can be applied to a generality of cases, except the structural analysis step. The structural analysis is tailored to a very specific type of bridge, hence diminishing the overall generality of the methodology. The manuscript should be then improved by splitting the structural analysis in a more general part, possibly suitable for a large variety of bridges including old constructions, and a more specific part needed for the present case study. Further improvement is needed in the analy-

sis of the impact on the road network, where too simplistic assumptions are made. In particular, it is implicitly assumed that no parts of the network other than the bridge are impacted by the flood, and that the capacity of the alternative routes are not limited. These assumptions may bring to a strong underestimation of the impact, neglecting the possibility of severe traffic jamming on the alternative routes and the need to take even longer re-routing due to the unserviceability of the nearest ones.

---

## Referee Comment (RC2) · Anonymous Referee #2 · 19 Feb 2021

The study combines a fluid dynamics model with a structural analysis model to assess the performance of an inundate bridge and then assess the impact on the functionality of the surrounding transport network. The paper lacks of basic bridge engineering understanding and bridge modelling and therefore has inaccuracies and provides limited insights for bridges exposed to flood hazard. The novelty is not clear, although hydrodynamic modelling is included to study the bridge response under flood effects, several simplifications are made, while the description of the models is not adequate. For example:

- The authors do not explain the loads used on the bridge. Are these code-based loads, which design situation/combination has been considered and for which elements?

- The loads shown in Fig. 2 have no relevance to bridge engineering, while there are

errors in bridge engineering terminology.

- The deformations shown in Fig. 2 for bearings are basic, described with wrong terminology and not relevant to the paper. Not correct bearing modelling/bearing failures are shown. Elastomeric bearings are usually deteriorated on isolated bridges, while their connections to the super/substructure are very critical and not discussed in the paper at all (a contact-like connection is insinuated through friction). Instead based on line 280 of the manuscript "These bearing elements were connected to rigid links, which simulated cap beams. . ." i.e. a fully rigid connection. Furthermore, it is not clear if uplift of the deck from the isolators is modelled in Opensees, and if this was done it should be further explained.

- Figure 4 indicates 'abutment', but no abutment is shown here.

- Foundation is shown fully fixed. This is not an acceptable assumption especially for a river crossing bridge. Foundation and SSI effects are activated under dynamic loads, like flooding.

- Yielding of the girders or piers is considered (line 125), however, it is not clear if nonlinearities of the bridge elements were considered in the model, and if this is the case then it is not sufficiently explained; for example, how the nonlinearity of the deck or reinforcement has been included in the model.

- Opensees is an advanced software to simulate the performance of structural systems subjected to earthquakes. It is not clear, why this software was selected for an oversimplified bridge model, and in particular, if a linear elastic simulation has been adopted. Also, validation of the models is not provided.

- The bridge deck and girders are modelled as rigid bodies, however, this is an oversimplified approach. Also, shear, flexural, and axial stiffness properties of the deck are not provided. In line 195, it is mentioned that "The bridge deck and girders are modelled as a rigid cross section (i.e. in 2D)"; this is confusing as a 3D model is shown in Figure

4.

- It is not clear if the CFD model accounts for the river-bed and river channel characteristics, the model input is not explained sufficiently, and no information of the model are given. It is not clear what is the ouput of the hydrodynamic model (e.g. time-histories of the hydrodynamic force?) and how then this output is imposed in the Opensees model.

- In section 2.3, the authors provide relevant literature for the definition of slight, moderate, extensive, and complete damage states, however, it is not clear which thresholds values have been used for the damage assessment of each bridge component, e.g. piers, bearings, deck in section 3.1.

- The framework includes a reliability analysis; however, no such analysis is conducted, which by definition is based on failure probabilities of the structure under study.

- The horizontal lines in Fig. 5 are not defined in the legend or figure caption.

- Recent papers that study the vulnerability of bridges to flood effects are not included in the literature review, e.g.:

Kim, H., Sim, S. H., Lee, J., Lee, Y. J., & Kim, J. M. (2017). Flood fragility analysis for bridges with multiple failure modes. Advances in Mechanical Engineering, 9(3), 1687814017696415.

Ahamed, T., Duan, J. G., & Jo, H. (2020). Flood-fragility analysis of instream bridges–consideration of flow hydraulics, geotechnical uncertainties, and variable scour depth. Structure and Infrastructure Engineering, 1-14.

Hung, C. C., & Yau, W. G. (2017). Vulnerability evaluation of scoured bridges under floods. Engineering Structures, 132, 288-299.

---

## Author Comment (AC1) · 5 Apr 2021

Please see attachment

Please also note the supplement to this comment:
https://nhess.copernicus.org/preprints/nhess-2020-375/nhess-2020-375-AC1-supplement.pdf

---

## Author Comment (AC2) · 5 Apr 2021

**Assessing flooding impact to riverine bridges: an integrated analysis**

Maria Pregnolato[1*,] Andrew O. Winter[2], Dakota Mascarenas[2], Andrew D. Sen[3], Paul Bates[4], Michael R. Motley[2]

[1] Dep. of Civil Engineering, University of Bristol, Bristol, BS8 1TR, UK
[2] Dep. of Civil and Environmental Engineering, University of Washington, Seattle, 98103, USA
[3] Dep. of Civil, Construction and Environmental Engineering, Marquette University, Milwaukee, 53233, USA
[4] School of Geographical Sciences, University of Bristol, Bristol, BS8 1RL, UK

*Correspondence to*: Maria Pregnolato (maria.pregnolato@bristol.ac.uk)

| No. | Comment | Answer |
|---|---|---|
| **Reviewer #1** | | |
| **1.1** | The manuscript presents an integrated framework for the assessment of flood impact on riverine bridges and the road network they connect, with the merit of including all the most relevant aspects of the problem, from the hydraulic to the structural and road network ones. | Authors appreciated the merit and complexity of the paper is highlighted by the reviewer. |
| **1.2** | The methodology is demonstrated on a UK case study, and it is intended to be applicable also elsewhere. Most of the various steps in the methodology are based on approaches that can be applied to a generality of cases, except the structural analysis step. | Authors appreciated the transferability of the method is highlighted by the reviewer. The structural analysis step will be made more general as suggested (see **Comment no. 1.3**). |
| **1.3** | The structural analysis is tailored to a very specific type of bridge, hence diminishing the overall generality of the methodology. The manuscript should be then improved by splitting the structural analysis in a more general part, possibly suitable for a large variety of bridges including old constructions, and a more specific part needed for the present case study. | The authors agree that the generality of the methodology should be preserved in the manuscript. To help clarify the distinction between the methodology and the case study, the text in Sec. 2.3 will be generalized to consider other types of bridges and the specific analysis details will be limited to Sec. 3.1. |
| **1.4** | Further improvement is needed in the analysis of the impact on the road network, where too simplistic assumptions are made. In particular, it is implicitly assumed that no parts of the network other than the bridge are impacted by the flood, and that the capacity of the alternative routes are not limited. These assumptions may bring to a strong underestimation of the impact, neglecting the possibility of severe traffic jamming on the alternative routes and the need to take even longer re-routing due to the unserviceability of the nearest ones. | We agree with the reviewer that assuming that no parts of the network other than the bridge are impacted by the flood (thus have reduced capacity) is an important simplification. We were aware of it, as stated in Sec. 4 (L379-382). This assumption is based on the following motivations. 1) This study is highly complex and multi-disciplinary; as such, its components needed to be kept at a low-complexity level in order to explore the combination of them (first of its kind). Nevertheless, all simplifications are |

| | | | clearly stated, and taken as points for future development (Sec. 4).

2) This study focused on riverine flooding, whose hazard footprint mainly affects bridges (Fig. 3b). For properly analysing the flooding impact to road networks, simulation of surface water flooding should be undertaken; this analysis would be a study on its own, and currently out of the scope of this piece of research. This will be better specified in Section 4.

3) This assumption implies an underestimation of the impact, and this point will be made clear. As stated in Sec. 4 (L373 and L383-4), the produced outputs are conceptual results and the importance of this work resides in the proof of concept of a new holistic methodology, rather than the quantitative results. |
|---|---|---|---|
| **Reviewer #2** | | | |
| **2.1** | The study combines a fluid dynamics model with a structural analysis model to assess the performance of an inundate bridge and then assess the impact on the functionality of the surrounding transport network. | | - |
| **2.2** | The paper lacks of basic bridge engineering understanding and bridge modelling and therefore has inaccuracies and provides limited insights for bridges exposed to flood hazard. The novelty is not clear, although hydrodynamic modelling is included to study the bridge response under flood effects, several simplifications are made, while the description of the models is not adequate. | | The authors appreciate the reviewer's expertise and insight in bridge engineering and will modify the manuscript to improve word use to ensure clarity. The novelty of the paper is the presentation of an interdisciplinary approach to evaluate performance of bridges subjected to flood loading and potential disruptions to transportation networks. In particular, very few studies have attempted to closely link computational fluid dynamics and structural analysis models.

The reviewer suggests that the models employed in the present evaluation are inaccurate and provide limited insight. To be clear, the relatively simple case study models reflect the limited knowledge of the actual bridge and site conditions, and the authors are cognizant of the uncertainty in actual bridge performance due to the potential for other hydrodynamic conditions, structural limit states, or transportation network vulnerabilities that are excluded from the study. With these issues in mind, the authors will improve the descriptions of each of the |

| | | models and identify specific limitations of the approaches that may be addressed in future work. |
|---|---|---|
| 2.3 | The authors do not explain the loads used on the bridge. Are these code-based loads, which design situation/combination has been considered and for which elements? | Since the paper describes a general methodology for evaluating performance under flood loading, a specific bridge design code or standard is not used in this study. Moreover, the loading considered here applies to structures with inadequate freeboard and/or structures subject to orifice flow and inundation. Flow characteristics for such conditions may be well established in the literature, but forcing on the superstructure is not. Therefore, the bridge loads are determined directly from the computational fluid dynamics model of the superstructure considering different combinations of flow heights and velocities that are possible at the site. The forces and moments on the bridge superstructure based on the computational fluid dynamics model results are the loads indicated in Fig. 2. These forces and moments are applied as distributed loads over the length of each girder and then transmitted to the other structural components (bearings, piers, etc.). An improved description of this procedure will be added to the manuscript, and the authors will also note how the methodology can be applied. |
| 2.4 | The loads shown in Fig. 2 have no relevance to bridge engineering, while there are errors in bridge engineering terminology.
 | These loads on the superstructure are relevant for high water flow leading to orifice flow conditions or inundation of the deck; consideration of these types of loads is not unique to this study (e.g., see Mondoro and Frangopol 2018). |
| 2.5 | The deformations shown in Fig. 2 for bearings are basic, described with wrong terminology and not relevant to the paper. Not correct bearing modelling/bearing failures are shown. Elastomeric bearings are usually deteriorated on isolated bridges, while their connections to the super/substructure are very critical and not discussed in the paper at all (a contact-like connection is insinuated through friction). Instead based on line 280 of the manuscript "These bearing elements were connected to rigid links, which simulated cap beams: : :" i.e. a fully rigid connection. Furthermore, it is | The authors disagree that these bearing deformations are irrelevant in view of the large forces modeled on the bridge superstructure. Complex nonlinear modeling of the bearing response including deterioration due to environmental exposure is not performed, but this is not the focus of the paper (the focus is on the methodology that links computational fluid dynamics, structural analysis, and transportation network evaluation).

The reviewer refers to L280 of the manuscript to suggest that the bearings in |

| | | |
|---|---|---|
| | not clear if uplift of the deck from the isolators is modelled in Opensees, and if this was done it should be further explained. | the model are modeled as rigid elements; however, the bearings are not modeled as rigid elements. "Rigid links" on this line refers to the condition between the cap beam and pier; this will be clarified in the revised manuscript. It is also understood that this is still a modeling simplification, and this will be noted. Uplift of the deck was not modeled in *OpenSees*, since this would imply significant movement of the superstructure that would invalidate the computational fluid dynamics model; instead, uplift and unseating (overcoming the assumed frictional resistance between the girder and bearing) is evaluated in post-processing. |
| 2.6 | Figure 4 indicates 'abutment', but no abutment is shown here.  | The abutments are not explicitly modeled; rather, the abutment locations are used as boundary conditions in the model. Figure 4 will be modified to clarify. |
| 2.7 | Foundation is shown fully fixed. This is not an acceptable assumption especially for a river crossing bridge. Foundation and SSI effects are activated under dynamic loads, like flooding. | This study focuses on the performance of bridge superstructures subjected to high water flow; therefore, soil-structure-interaction (SSI) has not been modeled. The authors are aware of this limitation, as stated in Sec. 3.1 (L323), and agree that detailed investigation of the foundation and SSI under this loading would be of significant interest. However, it is beyond the scope of the present paper. |
| 2.8 | Yielding of the girders or piers is considered (line 125), however, it is not clear if non-linearities of the bridge elements were considered in the model, and if this is the case then it is not sufficiently explained; for example, how the nonlinearity of the deck or reinforcement has been included in the model. | The manuscript will be updated to expand the description of the finite-element model. The girders and piers were modeled as fiber-based line elements with nonlinear constitutive models for the concrete and reinforcing steel within their respective cross sections. In addition, geometric nonlinearity was modeled (P-Delta effects), though significant influence of geometric nonlinearity would imply deformation that invalidates the computational fluid dynamics model (this was not observed). |
| 2.9 | Opensees is an advanced software to simulate the performance of structural systems subjected to earthquakes. It is not clear, why this software was selected for an oversimplified bridge model, and in particular, if a linear elastic simulation has | The authors suggest that a multitude of different software packages may be able to adequately characterize flood loading on the superstructure. The authors have never suggested that *OpenSees* is the only suitable option structural analysis within this methodology. However, this software was |

| | | |
|---|---|---|
| | been adopted. Also, validation of the models is not provided. | selected for use in the case study in order to simulate potential nonlinear response in the girders and piers under the hydraulic loading. Alternative software will be mentioned as possible options in Sec. 2.

The reviewer's observation that no validation of the models is provided is well-taken. The authors recognize that this is limitation of the study, however, there is a dearth of data available to support validation. The authors note that the methods employed by the authors have been used in prior studies. |
| 2.10 | The bridge deck and girders are modelled as rigid bodies, however, this is an oversimplified approach. Also, shear, flexural, and axial stiffness properties of the deck are not provided. In line 195, it is mentioned that "The bridge deck and girders are modelled as a rigid cross section (i.e. in 2D)"; this is confusing as a 3D model is shown in Figure 4. | The authors agree that the original language is too confusing and will clarify that the bridge superstructure is rigid in the computational fluid dynamics model (an important simplification to ensure the feasibility of the study) but not in the finite-element model; to that end, the authors can also provide the stiffness parameters from the finite-element model. Figure 4 shows the finite-element model only. |
| 2.11 | It is not clear if the CFD model accounts for the river-bed and river channel characteristics, the model input is not explained sufficiently, and no information of the model are given. It is not clear what is the output of the hydrodynamic model (e.g. time-histories of the hydrodynamic force?) and how then this output is imposed in the Opensees model. | The CFD model assumes a constant value of the section, calculated as possible from available data. Model input are listed in Table 2 and explained in Sec. 3 (L260-272).

The output of the hydrodynamic model includes water velocity and depth (1-in-a-500-year flood event). The values were extracted in proximity of the bridge, and also validated with historical data and inspection reports. The statistics for the velocity (both in its actual flood flow direction and also normal to the bridge) were computed from the LISFLOOD-LP velocity vectors Vx/Vy data and the maximum water depth, for both considering maximum values over the whole flood simulation. These values provide the initial data for the CFD model; in fact, the OpenFOAM model was set to simulate a range of flow velocity and depth values above and below the calculated 500-year flood results in order to assess how varying the depth and velocity affected the resulting bridge performance (see L260-8). The 500-year return period flood showed velocity values up to roughly 3.5 m/s and max flood depth up to 17 m near the M6 Bridge. These statistics motivates using a range of steady- |

| | | | |
|---|---|---|---|
| | | | state velocities of 1-3 m/s and inundation heights of 12.5-18 m above datum (14.8 m above river bottom) respectively, with the bottom of the bridge's lowest girders at approximately 12.375 m and the top at 14.425 m above +3.2 m datum. |
| **2.12** | In section 2.3, the authors provide relevant literature for the definition of slight, moderate, extensive, and complete damage states, however, it is not clear which thresholds values have been used for the damage assessment of each bridge component, e.g. piers, bearings, deck in section 3.1. | | The authors have inferred damage states based on the most probable failure modes examined in Fig. 5. The manuscript will be revised to explicitly relate the analysis results to the damage states. The connection between Sec. 2.3 and Sec. 3.1 will be better defined, by connecting Sec.3.1 and Fig.5, with Table 1. |
| **2.13** | The framework includes a reliability analysis; however, no such analysis is conducted, which by definition is based on failure probabilities of the structure under study. | | We agree with this note and thank the reviewer for the comment. It is correct that no statistical distribution of flood intensity was considered, therefore the term "reliability analysis" was substitute by "performance evaluation". |
| **2.14** | The horizontal lines in Fig. 5 are not defined in the legend or figure caption.  | | Figure 5 will be amended to indicate the limit state thresholds plotted in grey. Relevant equations from the literature will be included to the manuscript to show how these values are computed. |
| **2.15** | Recent papers that study the vulnerability of bridges to flood effects are not included in the literature review, e.g.:
• Kim, H., Sim, S. H., Lee, J., Lee, Y. J., & Kim, J. M. (2017). Flood fragility analysis for bridges with multiple failure modes. Advances in Mechanical Engineering, 9(3), 1687814017696415.
• Ahamed, T., Duan, J. G., & Jo, H. (2020). Flood-fragility analysis of instream bridges–consideration of flow hydraulics, geotechnical uncertainties, | | These references have been reviewed and will be included in the paper. |

| | and variable scour depth. Structure and Infrastructure Engineering, 1-14.
• Hung, C. C., & Yau, W. G. (2017). Vulnerability evaluation of scoured bridges under floods. Engineering Structures, 132, 288-299. | |
|---|---|---|

---

## Author Response (AR2)

**Assessing flooding impact to riverine bridges: an integrated analysis**

Maria Pregnolato[1*], Andrew O. Winter[2], Dakota Mascarenas[2], Andrew D. Sen[3], Paul Bates[4], Michael R. Motley[2]

[1] Dep. of Civil Engineering, University of Bristol, Bristol, BS8 1TR, UK
[2] Dep. of Civil and Environmental Engineering, University of Washington, Seattle, 98103, USA
[3] Dep. of Civil, Construction and Environmental Engineering, Marquette University, Milwaukee, 53233, USA
[4] School of Geographical Sciences, University of Bristol, Bristol, BS8 1RL, UK

*Correspondence to*: Maria Pregnolato (maria.pregnolato@bristol.ac.uk)

| No. | Comment | Answer |
|-----|---------|--------|
| **Reviewer #1** | | |
| 1.1 | The paper deals with a topic of extreme interest, and correctly identifies gaps in the current literature. However, probably due to its ambition to define "a rigorous, multiphysics modelling approach for the hydrodynamic forces impacting inundated bridges, and the subsequent structural response, while understanding of the consequences of such impact on the surrounding network", it fails in its objectives and to the Reviewer's opinion appears more like a loose combination of various tools that are not well integrated together, with a not very rigorous application to a case study. | Authors appreciated that the Reviewer highlights the relevance of the topic, as well as the complexity of the work. Since the study is complex indeed, the modeling approach presented herein used several intentional simplifications for demonstration purposes, including reducing the CFD domain or neglecting soil-foundation effects, and assumed rigidity of the structural system among others. In scenarios where these issues (or others) may be of more concern for a particular bridge, the fidelity of the modeling approach could be improved. Additionally, the failure states presented here may not translate broadly to the general bridge inventory, but additional or alternative structural/functional failure states could be applied. We are aware of the assumptions and associated limitations are discussed (L426-443). Nevertheless, the proposed approach of impact analysis can give practitioners a holistic method for assessing susceptibility to flooding and relative consequences at systemic level. The case study presented here represents an archetype for this approach. Further, there has been a trend in recent years to develop interdisciplinary approaches to larger-scale engineering problems such as those presented herein. This approach provides a framework for engineers and researchers to study a problem by taking advantage of seemingly disparate physical models that address the whole problem, without addressing its various sub-problems in silos. |
| 1.2 | pag.7 line 211. The limitations of the proposed approach (no interaction between | Thank you for this comment. For many large-scale systems subject to natural hazards, full |

| | | CFD and FE models) are not discussed. Can the bridge considered as case study be assumed as rigid for the purpose of evaluating the hydrodynamic forces, even in the case of such tall piers?

Why are the piers not modeled in the computational fluid dynamic analyses? | coupling of the CFD and FE models is impractical. While we agree that the structure will interact with water, for a civil structure we need to require small enough displacements to maintain general serviceability. Thus, we feel confident that the rigid model is (a) acceptable given the assumption of small displacements, and (b) important to show the loads that will be present on the structure if it remains in place as designed, which is the ultimate objective of this type of study. We did not model the substructure for a related reason, since we are examining the loading on the bridge superstructure, and within the rigid assumption we would not expect the inclusion of any piers to have a notable effect on these values, since the critical loads for the superstructure would be at midspan between the piers where their influence on the fluid flow would be negligible. The fluid forces would be larger away from the piers as well since the flow would not be obstructed by the pier, resulting in larger flow velocities. Using the loads computed on a midspan section of the bridge as distributed demands long the length of the bridge spans will give conservative results. |
|---|---|---|---|
| **1.3** | pag.8 line 247. In correspondence of the M6 bridge over Carlisle, there is a large floodplain with ample opportunities for creating additional discharge. It seems very unreasonable that even under a rare-high intensity flood will be inundated.

In fact, the floodwater maximum depth for a 500yrs return period is equal to 8.424m according to Figure 3. This suggests that the deck won't be inundated.

Please consider a more realistic case study, for example one of the many masonry arch bridges that are more likely to be severely affected by floods. | The city of Carlisle was chosen because it is a flood-prone area. We are aware that the actual inundation depths may be associated with extreme return periods; these were selected to ensure that the bridge was actually inundated in the OpenFOAM CFD models. Regarding this aspect, it is important to emphasize that this work focuses on demonstrating a bridge risk assessment workflow using an arbitrarily-selected bridge for which we were able to obtain design data and create a structural model using OpenSees - compared for example to historical bridges in downtown Carlisle (e.g., masonry bridges that would require more advanced FEM modelling methods to assess failure). The M6 bridge was used as a proof of concept, and the choice was dictated by available data. Authors agree that other bridges could provide results for flooding events with a lesser return period, that would be more likely to occur on a regular basis. Nevertheless, the significance of the |

| | | | |
|---|---|---|---|
| | | | methodology does not change. We are aware that the inundation depths we simulated are unlikely to actually occur (although climate change impact is very uncertain), indeed the main focus of the work is the risk assessment workflow. Ultimately, the approach presented here should be independent of a specific bridge or location. We used a bridge for which we managed to collect the most available data. At the moment, we are looking to gather data and structural details of other bridges in the area, with the view of analysing "combined" consequences. This aspect was clarified in the text (451-452). |
| **1.4** | pag. 11 line 303. The values of the lateral forces, vertical forces, and roll moments determined from OpenFOAM are not reported. Similarly, many information in the rest of the paper are missing. This is not good practice in scientific paper writing, since a journal paper should report enough data to allow results to be checked and also reproduced by others. | | We appreciate that more details could have been provided, as noted by the Reviewer. Figure 6 was added, as a "counterpart" of Fig. 7 (former Fig. 6), to show examples ofconverged outputs from the discussed simulations for initial flow velocity equal to 3 meters per second for all individual components and initial inundation levels. In addition to providing the requested results, a description of the OpenFOAM solvers and models that were used for all simulations as well as related physical parameters have been described in the text (L277-283). Girder profile geometry were also added as a part of Fig. 5, with some additional superstructure dimensions too (e.g. superstructure width). These modifications to the text along with the bridge geometry description, as well as the initial and boundary condition data provided in the OpenFOAM model description given in Section 3, give sufficient data to replicate the CFD analysis results. |
| | pag. 13 Line 351. If the piers and abutments are founded on rock, why is scour analysis carried out? Nevertheless, scour depths of 1-2m for foundations on piles are not a big problem. | | We thank you for this note and we agree. The text has been modified so that scour is discussed as a general issue relevant for many bridges in Section 2 (L61-63), but excluded from Section 3 (L319-320). |
| | line 325-332. The coefficient of friction between rubber and concrete is more likely to be higher than 0.4, and the assumption of a value of 0.1 seems unjustified for the assessment. Similarly, assuming that the dowels do not provide any shear resistance is too conservative. | | The authors agree in principle with the Reviewer and state on lines 352-355 that the assumed coefficient of friction and lack of dowel resistance is indeed conservative. The coefficient of friction value of 0.1 was selected to illustrate how the data may be used to evaluate bridge performance, not to suggest that this is the actual coefficient of |

| | | | |
|---|---|---|---|
| | | friction. The text has been updated as follows to clarify how this is used in the evaluation: "To illustrate how the sequential fluid-structure modeling results may be applied, a highly conservative, reduced coefficient friction of 0.1 is considered. Using this threshold, the results indicate flow conditions for which the given frictional resistance is approached or exceeded: 13.5-m flow depth with velocity of at least 6 m/s, 15-m flow depth with velocity of at least 5 m/s, 18-m flow depth with velocity of at least 6 m/s." | |
| | Why are bridge geometry and characteristics considered as exposure when they control the vulnerability and they even affect the hydraulic hazard? | The concepts of hazard, vulnerability and exposure can have slightly different interpretation. The following definition are considered for this study (from Grossi and Kunreuther, 2005). Hazard: possible, future occurrence of natural or human-induced physical events that may have adverse effects on vulnerable and exposed elements. Exposure: characteristics of the "asset at risk", *i.e.* an object at risk of damage or a business/service at risk of interruption (location, material, etc.). Vulnerability: the susceptibility to damage of elements, or other forms of loss, because of the hazard impact that can express via relationship e.g. damage curves. An application of vulnerability and criticality assessment is in Johnson and Whittington (2011). Exposed elements become vulnerable in the presence of the hazard(s) only.

Therefore, the components of the bridge (and of the road network) themselves are considered exposed elements, i.e. exposure. | |
| | Line 257. Does "Pier width" denote "Pier height"? | Pier width does not refer to the height of the pier columns. The intent was to describe the out-to-out width of each bridge deck for each direction of traffic. Since the original naming of this parameter was misleading, it will be changed to an alternate terminology such as superstructure (or bridge bent width). The width of the superstructure is 17.3 m (Table 2). | |
| | Improve caption of Figure 6. | The caption has been improved as below to specify what each subfigure shows. | |

| | | "Maximum simulated demand on elastomeric bearings in M6 bridge, including (a)/(b) shear force, (c)/(d) total shear strain due to combined axial, moment, and shear demands, and (e)/(f) shear deformation; plots on left show demand versus flow velocity and plots on right show demand versus flow height." |

**Assessing flooding impact to riverine bridges: an integrated analysis**

Maria Pregnolato[1*,] Andrew O. Winter[2], Dakota Mascarenas[2], Andrew D. Sen[3], Paul Bates[4], Michael R. Motley[2]

[1] Dep. of Civil Engineering, University of Bristol, Bristol, BS8 1TR, UK
[2] Dep. of Civil and Environmental Engineering, University of Washington, Seattle, 98103, USA
[3] Dep. of Civil, Construction and Environmental Engineering, Marquette University, Milwaukee, 53233, USA
[4] School of Geographical Sciences, University of Bristol, Bristol, BS8 1RL, UK

*Correspondence to*: Maria Pregnolato (maria.pregnolato@bristol.ac.uk)

| No. | Comment | Answer |
|---|---|---|
| **Reviewer #2** | | |
| **2.1** | This is an interesting, very well written and organized paper. I liked the combination of detailed modeling with existing qualitative and quantitative data. The work as presented seems to have a lot of potential for expansion into a larger network and as the basis for difficult decisions. | The authors thank the reviewer for this appreciation. |
| **2.2** | I have grown weary of seeing the Wardhana and Hadipriono, 2003, in papers. It is now nearly 20 years since it's publication and that was based on older data. Much has changed since then and I question whether it is in any way representative of the current situation. It would be worthwhile de-emphasizing its relationship to current situations. | Unfortunately, Wardhana and Hadipriono (2003) remains the study of reference for statistics of bridge failure (analysis of 500 failures in 1989-2000), although we agree their results could not be representative of our times. Nevertheless, scour is one of the major challenge and damage cause to bridges. The sentence has been modified as follow, with newer references (avoiding the direct reference to the "50%"): "Flood and scour represent one of the most frequent causes of bridge failures (Hunt, 2009; Wardhana and Hadipriono, 2003; Khan, 2015; Ahamed et al., 2020)" (L36). |
| **2.3** | There was a paper published 10 years ago that the authors might want to look at prior to publication, as it has some similar aspects, but a much simpler (maybe cruder?) approach. *Johnson, P.A., and Whittington, R.M., 2011. Vulnerability-based risk assessment for stream instability at bridges. Journal of Hydraulic Engineering, 137(10), 1248-1256.* | The work has been reviewed and considered in the literature review of the paper (L104-106). |

---

## Author Response (AR3)

**Assessing flooding impact to riverine bridges: an integrated analysis**

Maria Pregnolato[1*,] Andrew O. Winter[2], Dakota Mascarenas[2], Andrew D. Sen[3], Paul Bates[4], Michael R. Motley[2]

[1] Dep. of Civil Engineering, University of Bristol, Bristol, BS8 1TR, UK
[2] Dep. of Civil and Environmental Engineering, University of Washington, Seattle, 98103, USA
[3] Dep. of Civil, Construction and Environmental Engineering, Marquette University, Milwaukee, 53233, USA
[4] School of Geographical Sciences, University of Bristol, Bristol, BS8 1RL, UK

*Correspondence to*: Maria Pregnolato (maria.pregnolato@bristol.ac.uk)

We would like to thank once again the reviewers for their efforts and comments. Our hope is that we have sufficiently addressed these concerns for both the editorial team and the reviewers, and that we can move forward towards publication. It seems that at this stage the primary concern relates to the clarity of the objectives of this paper, which, from our standpoint, are to present a novel, multidisciplinary, multiphysics approach that considers the effects of a flooding hazard on a bridge and how that may impact the transportation network. This paper is objectively not a study of how bridge structures would generally respond to an extreme hazard, but how one may consider a local hazard in more detail and extend the analysis to the network at large. We feel that, at this point after four reviews, we have provided sufficient revision to this work to meet these objectives, and hope that we have resolved this disconnect. Further editorial changes related to specific physical response of specific individual bridge will not affect the primary objectives. Should a user want to refine their fluid modeling, structural modeling, or network modeling, that fits perfectly within the scope of what we present. Again, we thank the reviewer for their comments, which substantially improved the paper; and we hope that the editor can find that this paper is now suitable for publication.

| No. | Comment | Answer |
|-----|---------|--------|
| **Reviewer #3 (reject)** | | |
| **1.1** | *The scientific contribution of the paper is still unclear.*
 *If the contribution is the development of a holistic framework for assessing susceptibility to flooding and relative consequences at systemic level, as discussed in response to the Reviewer's comments, then such a framework was already developed by others (see below a list of few).*
 *If the novel contribution is the analysis of a realistic case study, then there are too many simplifications and there is no link between hydrological and hydraulic analysis results and too many physically inconsistent assumptions in the fragility analysis.*
 *The authors could select one of these topics (overarching framework, bridge fragility analysis under hydrodynamic loads, impact of floods on traffic redistribution, analysis of the resilience of transport infrastructure of Carlisle to flood hazard) and provide much more information on it, making clear what is the improvement of the paper with respect to the state of art.* | The main contribution of the paper is the development of a holistic framework for assessing susceptibility to flooding and relative consequences at systemic level (L16-18) using CFD (Computational Fluid Dynamics) and FE (finite element) modelling. The analysis of a realistic case study is not the main focus, rather an example of application of the framework and proof-of-concept (L20-22).
 Assessing the consequences associated with flooding requires an understanding of the hazard and how infrastructure might respond to them. Although there are many ways to do this assessment, we argue that our approach is novel because it models hydrodynamic forces as demand on the bridge structure using CFD. While there are, admittedly, several other frameworks that have been developed for similar cases, this concept is not ubiquitous throughout the literature, and expanded computing power has resulted in more availability of these tools, and our hope is that we are providing an avenue for potential users to explore these multi-hazard approaches.
 This work obviously builds on existing literature, but moves this forward since: *(i)* it develops a multiphysics, multilevel approach that takes advantage of seemingly disparate physical models, never integrated before; *(ii)* it represents a first attempt to couple CFD analysis with both Finite Element (FE) and network analysis, in an effort to capture both the cause and effect of flooding (as parallel studies are doing for other hazards, e.g. Liu et al. (2021) for fire). The novelty of this work is grounded in how this fits within a multidisciplinary approach that could more broadly extend complex engineering analysis of hazard resilience into a practical network analysis.

 We would like also to specify differences from cited works. Gehl and D'Ayala (2018) presented a multi-hazard risk assessment using functionality loss curve; they did not include consequences at network level and applied the framework to an "ideal" bridge |

and a hypothetical road network only; no damage mechanism is specified, a bridge is considered closed when submerged. Kim et al. (2018)'s framework focused on traffic forecasting and related cost (no mention of direct costs); they adopted hydrological and hydraulic analysis to determine hazard information for 200/300/400-years return period. The work of Lamb et al. (2019) is about scour only and develops new fragility analysis; moreover, their consequences assessment was for railways bridges (in terms of passenger journey disruption); their study is the only UK-data-based work. Alabbad et al. (2021) proposed an interesting high-level framework where bridge closure is considered only (and based on comparing flood depth and deck height); no hydraulic or CFD modelling is included; it is also noted they use a 100- and 500-year return period as events.

Missing works have been added in the manuscript and we thank the reviewer for the suggestion.

The scope of this work is exactly not to adopt a silo-based approach (by selecting one topic), but to challenge the current *status-quo* with a holistic view of the matter. This is precisely the novel contribution. If we were keen to focus on one topic, we would have selected another journal (e.g., Eng. Structure for fragility analysis, Transportation Part B for traffic redistribution, etc.). Our choice of NHESS was about targeting an interdisciplinary journal interested on "natural hazards and their consequences", which embraces a "holistic Earth system science approach". NHESS serves a wide and diverse community of research scientists, practitioners, and decision makers (quite often "generalist" of the wider subject), which are looking to understand the potential practical consequences to infrastructure due to extreme flooding – in the remit of this study.

The broader community (and two reviewers) has already positively engaged with this work (thanks to NHESS discussion and conference papers on preliminary parts of this study, see e.g. citation in Eidsvig et al. (2021)), and I am

| | | | currently discussing further application with a consultancy company which would like to use CFD for risk assessment. In this light, we are convinced that the article is worth publication, and eventual shortcomings will be challenged by the community. |
|---|---|---|---|
| **1.2** | | *I respectfully disagree with the response to comment #2. The investigated bridge has very tall piers, and from simple calculations (just assume 2 kN/m2 of hydrodynamic load) one could immediately have an idea of the potential impact of hydrodynamic loads on the pier deflection and on the top displacement demand, which is strongly related to the performance of the bearings.* | The authors appreciate the reviewer's concern of the effects of fluid-structure interaction. The structural analysis results using the flood loads determined from the CFD analysis show that the lateral drift of the piers is sufficiently low: 0.053% of the pier height under only the most severe loading. Similarly, the maximum top rotation sustained by any of the piers is 0.00104 rad. The piers are tall, but they have substantial stiffness in the direction of flow that limits their deformation and the potential for significant fluid-structure interaction; therefore, the main vulnerability is judged to be in the superstructure for this particular bridge. It is noted that fluid-structure interaction may be important for other bridge geometries and flow conditions, but this again gets to the objectives of the paper, and it is quite common for these to include these and similar assumptions when modeling these types of problems. |
| **1.3** | | *The case study bridge has not been changed. Even under a rare event such as the one considered (500 years return period), it is not expected to be flooded. The flow velocity of 3m/s at the deck level seems high, but no actual hydraulic analysis was carried out. A more realistic scenario should be analysed.* | The case study is an application of the framework and proof-of-concept. It was selected because all required data were available (as opposed to other bridges of the area), thanks to our collaboration with UK Highways Agency (formerly Highways England). The velocity of 3m/s was based on hydrodynamic simulation for the event of 500-year return period. Due to climate change, flood return periods are dramatically decreases, and a 500-year flood in 2021 could become a 271.6-year flood in the 2050s (Orton et al., 2016). Thus, for the design of e.g. bridge piers, return periods up to 500-year will be more and more justified (Rashidi et al., 2021); recent works has used similar return periods too (Alabbad et al., 2021). Highways Agency is supportive of the overall framework and expressed interest in investigating consequences due to extreme events for the M6 bridge in Carlisle (support letters available). |

| | | |
|---|---|---|
| | | The use of a coupled hydrodynamic and CFD analysis is part of the novelty of the work (rather a hydraulic analysis), since CFD has been suggested as a more sophisticated technique to be used for modelling flow depth and velocities at sites (Bento et al., 2021). This insight, again, gets to the objectives of this work. |
| 1.4 | *Figure 6 has been added, without no explanation of the results, which are not very clear. The obtained results should be explained and discussed.* | Discussion of obtained results have been provided in further depth preceding Figure 6 (L308-355).
 Moreover, also Fig. 5 was improved. |
| 1.5 | *The assumption of a coefficient of friction of 0.1 is not justifiable. If there is uncertainty in the friction, a Monte Carlo simulation should be carried out to investigate the effect of this uncertainty, or at least a sensitivity study should be performed to investigate the effect of the assumption* | The present study is not probabilistic in nature and thus a Monte Carlo simulation is beyond the scope of the work. Moreover, the discussion and use of 0.1 as a coefficient of friction is noted to be purely illustrative and admittedly highly conservative in the text (L412-413). The authors do not suggest a coefficient of friction of 0.1 is the true coefficient of friction, but it is based on (1) the suggested AASHTO Commentary design coefficient of friction of 0.2 and (2) the expectation that the coefficient of friction may be lower than expected in wet/submerged conditions. Further, the limiting coefficient of friction is defined as the AASHTO commentary suggestion of 0.2 in Figures 7a and 7b. To the authors' knowledge, there are no experimental data available to help overcome the epistemic uncertainty associated with these conditions. |

Alabbad, Y., Mount, J., Campbell, A. M. and Demir, I. (2021). Assessment of transportation system disruption and accessibility to critical amenities during flooding: Iowa case study. Science of The Total Environment, 148476

Bento, A.M. Viseu, T., Pêgo and J.P. Couto, L. (2021). Experimental Characterization of the Flow Field around Oblong Bridge Piers. Fluids, 6, 370. https://doi.org/10.3390/fluids6110370

Gehl, P. and D'ayala, D. (2018). System loss assessment of bridge networks accounting for multi-hazard interactions. Structure and Infrastructure Engineering, 14(10), 1355-1371.

Eidsvig, U., Santamaría, M., Galvão, N., Tanasic, N., Piciullo, L., Hajdin, R., Nadim, F., Sousa, H.S. and Matos, J. (2021). Risk Assessment of Terrestrial Transportation Infrastructures Exposed to Extreme Events. Infrastructures 6(11): 163. https://doi.org/10.3390/infrastructures6110163

Kim, B., Shin, S. C. and Kim, D. Y. (2018). Scenario-based economic impact analysis for bridge closures due to flooding: A case study of North Gyeongsang Province, South Korea. Water, 10(8), 981.

Lamb, R., Garside, P., Pant, R. and Hall, J. W. (2019). A probabilistic model of the economic risk to Britain's railway network from bridge scour during floods. Risk Analysis, 39(11), 2457–2478. https://doi.org/10.1111/risa.13370

Liu, Z., Silva, C.J.G., Huang, Q., Hasemi, Y., Huang, Y. and Guo Z. (2021). Coupled CFD–FEM Simulation Methodology for Fire-Exposed Bridges. J. Bridge Eng., 26(10): 04021074. https://doi.org/10.1061/(ASCE)BE.1943-5592.0001770

Orton, P.M., Hall, T., Talke, S.A., Blumberg, A.F., Georgas, N. and Vinogradov, S. (2016). A validated tropical-extratropical flood hazard assessment for New York Harbor J. Geophys. Res. Oceans, 121: 8904–29

Rashid, M.M., Wahl, T. and Chamberset D.P. (2021). Environ. Res. Lett., 16: 024026

| | | |
|---|---|---|
| **Reviewer #4 (accept as it is)** | | |
| - | - | - |

---

## Author Response (AR4)

**Assessing flooding impact to riverine bridges: an integrated analysis**

Maria Pregnolato[1*], Andrew O. Winter[2], Dakota Mascarenas[2], Andrew D. Sen[3], Paul Bates[4], Michael R. Motley[2]

[1] Dep. of Civil Engineering, University of Bristol, Bristol, BS8 1TR, UK
[2] Dep. of Civil and Environmental Engineering, University of Washington, Seattle, 98103, USA
[3] Dep. of Civil, Construction and Environmental Engineering, Marquette University, Milwaukee, 53233, USA
[4] School of Geographical Sciences, University of Bristol, Bristol, BS8 1RL, UK

*Correspondence to*: Maria Pregnolato (maria.pregnolato@bristol.ac.uk)

We would like to thank once all the reviewers, the editor and the editorial team for their support in getting this work improved and published.

| No. | Comment | Answer |
|-----|---------|--------|
| **Reviewer #3 (minor revision)** | | |
| 1.1 | Several responses to my comments have been quite evasive, for example in response to comment 1.3 instead of addressing the concern regarding the values of the flow velocity and height that seem excessive and are not supported by hydraulic analysis, the authors say that the Highways Agency is "supportive of the overall framework and expressed interest in investigating consequences due to extreme events for the M6 bridge in Carlisle". | The answer to previous Comment 1.3 also included the test below: "The velocity of 3m/s was based on hydrodynamic simulation for the event of 500-year return period. Due to climate change, flood return periods are dramatically decreases, and a 500-year flood in 2021 could become a 271.6-year flood in the 2050s (Orton et al., 2016). Thus, for the design of e.g. bridge piers, return periods up to 500-year will be more and more justified (Rashidi et al., 2021); recent works has used similar return periods too (Alabbad et al., 2021)." |
| 1.2 | I agree that climate change will exacerbate the risk of bridge failure due to floods, but other case studies would have been more effective and more realistic. The friction coefficient between concrete and elastomer can be very high (otherwise car tyres would not be made of elastomer and would not function under wet conditions!) and I am not sure AASHTO would prescribe a value of 0.2. In any case, 0.1 is a value typically used as an upper bound for teflon (a material used for guaranteeing sliding). | The authors do not suggest a coefficient of friction of 0.1 is the true coefficient of friction, but it is based on (1) the suggested AASHTO Commentary design coefficient of friction of 0.2 and (2) the expectation that the coefficient of friction may be lower than expected in wet/submerged conditions. Further, the limiting coefficient of friction is defined as the AASHTO commentary suggestion of 0.2 in Figures 7a and 7b. To the authors' knowledge, there are no experimental data available to help overcome the epistemic uncertainty associated with these conditions. |
| 1.3 | I pointed out that if the scope of the paper is to present an "holistic framework", others already did, introducing similar levels of simplification. However, I don't think it is worth at this stage to continue to point out such issues when they are considered as minor by the authors, and thus are not addressed. Thus, I leave to the Editor's judgement to decide on the suitability of such a manuscript for publication. | The previous response to Comment 1.1 largely explained the differences from existing works and the novelty of this paper. In summary, we argue that our approach is novel because it models hydrodynamic forces as demand on the bridge structure using CFD. While there are, admittedly, several other frameworks that have been developed for similar cases, this concept is not ubiquitous throughout the literature, and expanded computing power has resulted in more availability of these tools, and our hope is that we are providing an avenue for potential users to explore these approaches. |

Alabbad, Y., Mount, J., Campbell, A. M. and Demir, I. (2021). Assessment of transportation system disruption and accessibility to critical amenities during flooding: Iowa case study. Science of The Total Environment, 148476

Orton, P.M., Hall, T., Talke, S.A., Blumberg, A.F., Georgas, N. and Vinogradov, S. (2016). A validated tropical-extratropical flood hazard assessment for New York Harbor J. Geophys. Res. Oceans, 121: 8904–29

Rashid, M.M., Wahl, T. and Chamberset D.P. (2021). Environ. Res. Lett., 16: 024026